# Dissecting the early steps of MLL induced leukaemogenic transformation using a mouse model of AML

Silvia Basilico[1,5], Xiaonan Wang [1,5], Alison Kennedy [1], Konstantinos Tzelepis [2,3], George Giotopoulos [1], Sarah J. Kinston[1], Pedro M. Quiros [2], Kim Wong [2], David J. Adams [2], Larissa S. Carnevalli [4], Brian J.P. Huntly [1], George S. Vassiliou [1,2], Fernando J. Calero-Nieto [1✉] & Berthold Göttgens [1✉]

Leukaemogenic mutations commonly disrupt cellular differentiation and/or enhance proliferation, thus perturbing the regulatory programs that control self-renewal and differentiation of stem and progenitor cells. Translocations involving the *Mll1* (*Kmt2a*) gene generate powerful oncogenic fusion proteins, predominantly affecting infant and paediatric AML and ALL patients. The early stages of leukaemogenic transformation are typically inaccessible from human patients and conventional mouse models. Here, we take advantage of cells conditionally blocked at the multipotent haematopoietic progenitor stage to develop a MLL-r model capturing early cellular and molecular consequences of MLL-ENL expression based on a clear clonal relationship between parental and leukaemic cells. Through a combination of scRNA-seq, ATAC-seq and genome-scale CRISPR-Cas9 screening, we identify pathways and genes likely to drive the early phases of leukaemogenesis. Finally, we demonstrate the broad utility of using matched parental and transformed cells for small molecule inhibitor studies by validating both previously known and other potential therapeutic targets.

[1] Wellcome and MRC Cambridge Stem Cell Institute and University of Cambridge Department of Haematology, Jeffrey Cheah Biomedical Centre, Puddicombe Way, Cambridge CB2 0AW, UK. [2] Wellcome Trust Sanger Institute, Hinxton, Cambridge CB10 1SA, UK. [3] Milner Therapeutics Institute, University of Cambridge, Jeffrey Cheah Biomedical Centre, Puddicombe Way, Cambridge CB2 0AW, UK. [4] Oncology, AstraZeneca, 1 Francis Crick Avenue, Cambridge CB2 0AA, UK. [5] These author contributed equally: Silvia Basilico, Xiaonan Wang. ✉email: fjc28@cam.ac.uk; bg200@cam.ac.uk

Chromosomal rearrangements involving the Mixed Lineage Leukaemia gene (MLL-r) cause more than 70% of infant leukaemias with either myeloid (AML) or lymphoid (ALL) immunophenotype[1,2]. MLL-r also occur in 10% of adult AML cases, and in therapy-related acute leukaemias (t-ALs)[3,4]. Several retroviral and non-retroviral mouse models bearing MLL fusion proteins have advanced our understanding of MLL-fusion-mediated leukaemogenesis. The first retroviral MLL-fusion leukaemia model[5] employed retroviral transduction into lineage depleted or c-Kit sorted mouse bone marrow haematopoietic stem/progenitor cells (HSPCs) followed by culture in methylcellulose and subsequent injection into immunodeficient mice. A key goal of non-retroviral mouse models has been to achieve fusion gene expression levels representative of the endogenous gene loci involved in the translocation events[6,7]. However, there has been substantial phenotypic variation between the various mouse models.

One of the main causes of inconsistencies comes from the differences in the target cells, where different fluorescence-activated cell sorting (FACS) strategies result in overlapping but not identical populations. The use of two different sorting strategies for HSCs for example resulted in reports that MLL-r can[8] or cannot[9] transform the target cells. When assayed at the single-cell level either functionally or by molecular profiling, all conventionally defined haematopoietic stem/progenitor populations display substantial heterogeneity[10–12]. Consequently, the exact nature of the parental cell that has been transformed in any of the traditional retroviral leukaemia models remains ill defined. Studies aiming to decipher the early stages of leukaemic transformation are therefore impeded, and there is no appropriate cell type that can be used as wild-type control to represent the starting cells when testing putative drug candidates. Importantly, the commonly used Lineage negative (Lin−, mouse) or CD34 positive (CD34+, human) control cells do not address these issues, because these populations are very heterogeneous, and include stem cells but also erythroid, myeloid and lymphoid progenitors[12,13].

Cell lines that are conditionally blocked at the stage of stem/progenitor and maintain intact differentiation potential represent an attractive approach for deriving defined and reproducible sources of HSPCs. Cell line models requiring cytokines for their in vitro growth have been particularly sought after, as cytokine dependence represents a key aspect of the normal physiology of primary HSPCs[14,15]. The LMPP-like Hoxb8-FL cell line[16] stands out because of its validated multilineage in vitro and in vivo differentiation capacity. Hoxb8-FL cells carry a glucocorticoid-controlled Hoxb8 transgene, and require Hoxb8 induction as well as externally supplied Flt3 ligand (Flt3L) for in vitro propagation (self-renewal condition). Withdrawal of Hoxb8 coupled with various cytokine combinations allows directed differentiation into both myeloid and lymphoid lineages from a clonally derived precursor cell.

Here we report the development and validation of a mouse model of MLL-ENL-driven AML starting from Hoxb8-FL cells, which recapitulate all key features of bone marrow-derived retroviral AML models both in vitro and in vivo. Unlike previous models, the exact nature of the target cell is known and accessible in our model, allowing for direct comparisons between different stages. We then use this model to identify transcriptional changes during early leukaemogenic transformation using both single-cell RNA-seq (scRNA-seq) and ATAC-seq approaches, followed by genome-wide CRISPR-Cas9 screens to identify genetic vulnerabilities specifically associated with the transformed, but not the parental cells. Integrated data analysis followed by small molecule-based functional validation identifies therapeutic targets including DNA damage response (DDR) and metabolic pathways.

## Results

### Development of a clonal mouse model of MLL-ENL-driven AML.

Highly purified HSPC populations are recognised to be heterogeneous[10,17]. Therefore, it is difficult to define a precise wild-type parental control in conventional retroviral transduction leukaemia models. To circumvent this problem, we devised a strategy based on the clonal mouse haematopoietic progenitor cell line Hoxb8-FL[16]. Cells were transduced with either MSCV-MLL-ENL-GFP (henceforth referred as ME-Parental cells) or control MSCV-GFP (henceforth referred as Parental cells) and were serially re-plated in methylcellulose (CFU) in the absence of Flt3L and β-estradiol but in the presence of interleukin-3 (IL-3), interleukin-6 (IL-6), stem cell factor (SCF) and erythropoietin (EPO). This step was followed by liquid culture, first in the presence of IL-3, IL-6 and SCF, then IL-3 and IL-6 and finally IL-3 alone. Leukaemic transforming potential in vivo was assessed by transplantation into lethally irradiated mice (Fig. 1a).

Only ME-Parental cells (transduced with the MLL-ENL virus) were able to generate serially re-plating colonies (Fig. 1b) with a morphology that was either compact or compact with a halo of differentiating cells (Fig. 1c), as previously described for conventional bone marrow progenitor transduction experiments[5]. Following three rounds of plating in methylcellulose, MLL-ENL-transduced cells were grown in liquid culture to generate IL-3-dependent cells (hereafter referred to as ME-Transformed) that were maintained for over a month, with continuous exponential growth and a doubling time of 24 h (Fig. 1d). When compared with the wild-type Hoxb8-FL cells, flow cytometric analysis of the ME-Transformed sample showed acquisition of the myeloid surface markers CD11b and Gr-1 and downregulation of c-Kit (Fig. 1e). Of note, ME-Transformed cells did not show expression of CD11c, MHC class II, B220 and F4/80, reminiscent of an immature myeloid differentiation stage (Supplementary Fig. 1a, b).

To validate the generated MLL-ENL model in vivo, we transplanted Parental cells (n = 5) or ME-Transformed cells (n = 5) into lethally irradiated mice, together with CD45.2 bone marrow donor-derived cells. Development of acute myeloid leukaemia (AML) was monitored via flow cytometry of the peripheral blood. All mice transplanted with the MLL-ENL-transduced cells developed AML within 75 days; while none of the parental mice developed disease up to 100 days after injection (Fig. 1f), confirmed by the absence of GFP+ cells in the peripheral blood, spleen and bone marrow (Supplementary Fig. 1c). Characteristic features of AML including splenomegaly and hepatomegaly were only observed in mice transplanted with ME-Transformed cells (Fig. 1g and Supplementary Fig. 1d), consistent with previous reports of bone marrow haematopoietic progenitor cells transduced with MLL-ENL as well as other MLL fusion genes[5,18].

To further understand the clonal relationship within our model, we characterised GFP+ cells obtained from three different animals at the time of culling using flow cytometry (Supplementary Fig. 1e) and performed exome sequencing of these cells together with Parental and ME-Transformed cells. No additional driver mutations were found in cells obtained from leukaemic animals (Supplementary Data 1). The in vitro and in vivo experiments therefore validate our MLL-ENL-transduced cells as a preleukaemic model for AML, facilitating access to the early stages of transformation and providing authentic parental control cells for molecular and cellular comparisons.

### Leukaemogenic program requires exiting self-renewal conditions.

To investigate the transcriptional consequences of MLL-ENL expression, we sorted single GFP+ Parental, ME-Parental

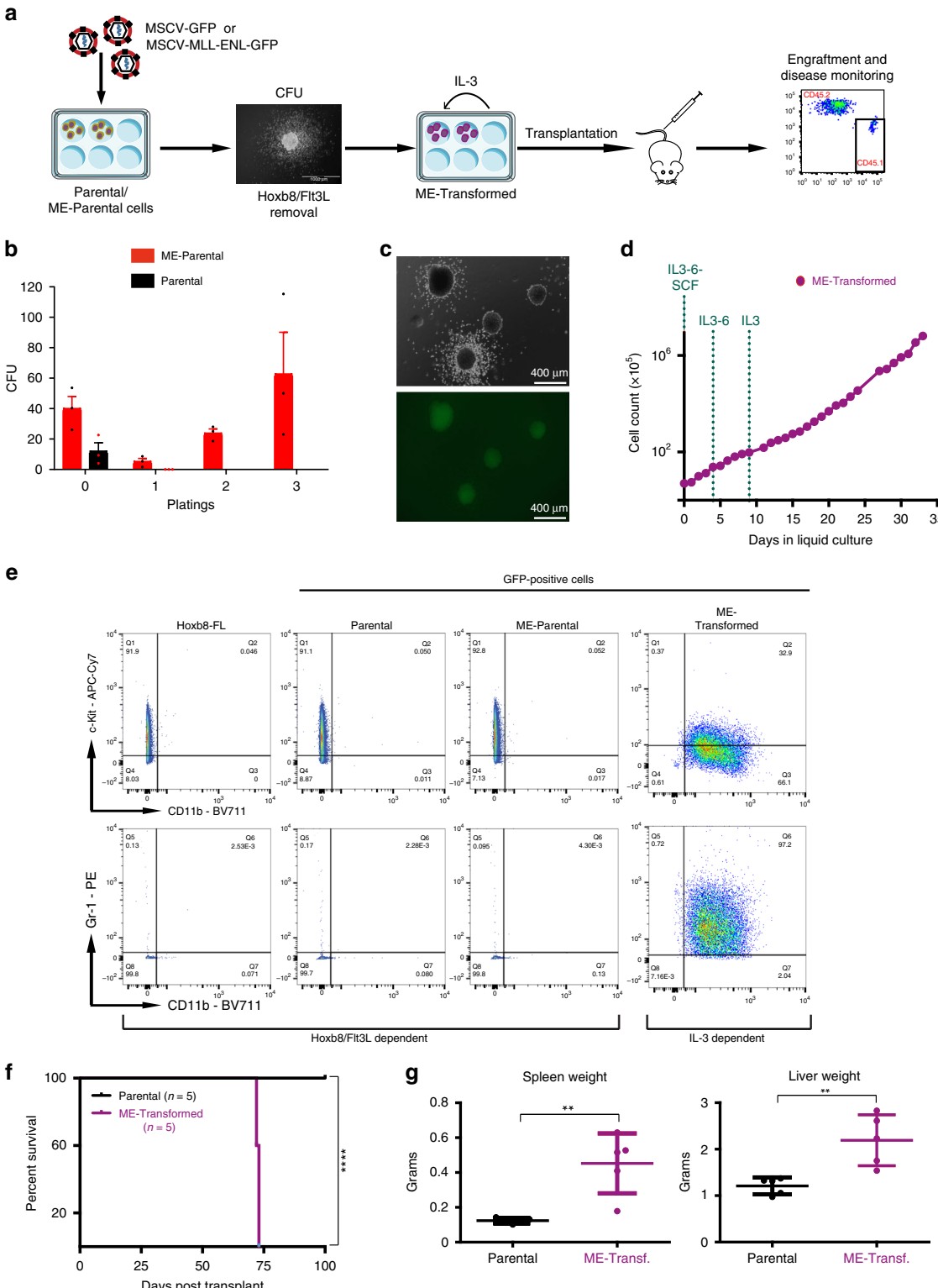

and ME-Transformed cells into 96-well plates for single-cell gene expression analysis (Fig. 2a). Conventional bone marrow progenitors transduced with MLL-ENL (hereafter referred to as MLL-ENL BM)[19] were included as positive controls. Single Parental and ME-Parental cells were processed for scRNA-seq using the Smart-Seq2 protocol[20]. ME-Transformed and MLL-ENL BM cells cultured in the presence of IL-3 were similarly profiled.

Principal component analysis (PCA) separated in the first component cells dependent on Flt3L and β-estradiol from IL-3-dependent transformed cells (Fig. 2b and Supplementary Fig. 2a). Analysis of the genes underlying this separation (PC1 loadings) revealed that Parental and ME-Parental cells expressed genes such as *Ddx4*, *Cd34* and *Ebf1* confirming the mixed lineage potential of Hoxb8-FL cells as described by Redecke et al.[16]. By contrast, both the MLL-ENL BM and ME-Transformed samples,

**Fig. 1 Development of a clonally derived MLL-ENL mouse model. a** Schematic outline of in vitro and in vivo experimental scheme. Hoxb8-FL cells were transduced with MLL-ENL or empty virus to generate ME-Parental and Parental cells, respectively. When cultured in methylcellulose without Flt3L and estradiol (causing retention of Hoxb8-ER in the cytoplasm), only ME-Parental cells had re-plating capabilities. These cells were transferred to liquid culture supplemented with IL-3, IL-6 and SCF. Sequentially, SCF and IL-6 were removed to obtain ME-Transformed cells able to grow long term in the presence of IL-3. These cells were then tail injected into lethally irradiated mice, followed by close monitoring of engraftment and disease onset. **b** Bar graph showing colony numbers at each round of methylcellulose plating (every 7 days). Mean and SEM are represented. $N = 3$ biologically independent experiments. **c** Colonies generated in methylcellulose by MLL-ENL-transduced Hoxb8-FL cells are characterised by either a compact centre or a compact centre with a halo of differentiating cells (images taken using an EVOS® inverted fluorescence microscope). This experiment was repeated three times with similar results. **d** Growth curve of ME-Transformed cells following dissociation of methylcellulose colonies obtained after three replatings. Cells were kept in liquid culture supplemented with IL-3, IL-6 and SCF for two passages. SCF was removed from the culture for the following two passages, IL-6 was then also removed after two passages and then cells were cultured only with IL-3. **e** Phenotypic characterisation of untransduced (Hoxb8-FL) and transduced (Parental and ME-Parental) cells cultured in the presence of Flt3L and β-estradiol compared to ME-Transformed cells cultured in IL-3-only medium. ME-Transformed cells showed upregulation of myeloid markers CD11b and Gr-1 and downregulation of c-Kit, in contrast to Hoxb8-FL, Parental and ME-Parental cells. **f** Survival curve of mice transplanted with either Parental cells ($n = 5$) or ME-Transformed cells ($n = 5$) (two-sided log-rank Mantel-Cox test. ****p value = 0.0031). **g** Graphs showing difference in spleen (**p = 0.0028) and liver (**p = 0.0052) weight between ME-Transformed and Parental mice. Average of five mice per condition ± SEM, two-tailed unpaired t test. Source data are provided as a Source Data file.

adapted to growth in IL-3, expressed myeloid lineage genes such as the neutrophil lineage marker *Elane*[21] and the granulocyte marker *Ly6c2* (*Gr-1*)[22] (Fig. 2c). Moreover, both samples expressed genes previously associated specifically with MLL-mediated leukaemic transformation such as the transcription factor Six1[18,23].

To identify the likely counterparts in normal haematopoiesis for the four populations profiled here, we projected the single-cell transcriptomes onto a force-directed graph representation of over 40,000 published single transcriptomes from normal bone marrow HSPCs[24] (Fig. 2d). Serving as an important positive control in this analysis, the Parental and ME-Parental cells cultured in Flt3L and β-estradiol mapped to the region that contained lymphoid and myeloid progenitors, consistent with their multipotent progenitor identity. In contrast, many of the MLL-ENL BM and ME-Transformed cells cultured in IL-3 mapped to more mature cells, clustering within the neutrophil and more mature monocytic branches of the single-cell transcriptional landscape.

The most surprising observation was the similarity of the Parental with the ME-Parental cells (as evidenced by PCA analysis and projections onto the HSPC landscape), suggesting that overexpression of MLL-ENL alone in the Hoxb8-FL cells is not sufficient to initiate a leukaemogenic transcriptional program. All four cell types studied here express key genes reported to be involved in MLL fusion-driven transformation (*Meis1* and *Hoxa9*)[25] and genes involved in proliferation and myeloid cell differentiation (*Myb* and *Myc*)[26,27] (Fig. 2e). To investigate further any potential impact of MLL-ENL expression on the Hoxb8-FL cells cultured in Flt3L and β-estradiol, we compared the transcriptional profile of ME-Parental cells to Parental cells. We found that both cell types are very homogenous and could not detect significant differences at the transcriptomic level between them (Supplementary Fig. 2b). We also examined the chromatin structure of ME-Parental cells in comparison to the Parental cells using ATAC-seq[28] given that MLL is recognised as an "epigenetic regulator" that can influence chromatin state. Three pools of 50,000 GFP+ cells were analysed at 6 and 9 days post transduction. Visual inspection of key MLL leukaemia-associated gene loci showed no significant differences in chromatin accessibility profiles at either time point (Supplementary Fig. 2c). Moreover, following peak calling, comparison of the coverage at all regions called as a peak in either Parental or ME-Parental samples showed no statistical differences at either day 6 or day 9 post infection (Fig. 2f, g) and displayed a similar genome-wide distribution (Supplementary Fig. 2d). Taken together, our results show that MLL-ENL can induce a leukaemic

transcriptional program in Hoxb8-FL cells, but only if the cells are taken out of their Flt3L and β-estradiol self-renewal culture condition, reminiscent of previous studies showing that MLL-ENL did not induce AML in mice when transduced into highly purified HSCs[9] and MLL-AF9 did not cause AML in either HSCs or CMPs when myeloid differentiation was compromised by C/EBPα deletion[29].

**Defects in cytokine-induced differentiation caused by MLL-ENL.** Previous studies indicated that AML development in the murine MLL-AF9 model required myeloid differentiation[29]. To capture early impacts of MLL-ENL on myeloid differentiation, we took Parental and ME-Parental cells out of the Flt3L and β-estradiol self-renewal conditions, and exposed them to one of three myeloid differentiation cytokines: IL-3, GM-CSF or Flt3L (Fig. 3a). Myeloid differentiation was assessed before cytokine addition (day 0) and after 4 and 7 days of stimulation (Fig. 3b and Supplementary Fig. 3a). Of note, all three cytokines resulted in downregulation of c-Kit expression consistent with loss of the immature LMPP-like phenotype of Hoxb8-FL (Fig. 3b).

Effects of MLL-ENL expression on myeloid maturation were already evident at day 4 for the IL-3 or GM-CSF treatments, and then also for Flt3L at day 7. An overall delay of myeloid differentiation was apparent, since ME-Parental cells were CD11b$^{-/low}$ in IL-3 or GM-CSF at day 4 and in Flt3L at day 7, whilst the majority of the Parental cells were CD11b$^{high}$ at the same time points. By day 7, the difference in myeloid maturation of ME-Parental cells compared to Parental was particularly large for both the IL-3 and GM-CSF treatments (Supplementary Fig. 3b). Following IL-3 exposure, 62.8% of MLL-ENL cells displayed a granulocyte phenotype being CD11b$^{low}$ Gr-1$^{+}$, with reduced levels of antigen-presenting cell markers (CD11c, MHC II and B220) and the macrophage marker F4/80 (Supplementary Fig. 3a). Parental cells, on the other hand, generated a mixture of more mature cells, such as macrophages (CD11b$^{+}$ Gr-1$^{-}$ MHCII$^{+}$ CD11c$^{+}$ B220$^{+}$ F4/80$^{+}$) and dendritic cells (CD11b$^{+}$ Gr-1$^{-}$ MHCII$^{+}$ CD11c$^{+}$ B220$^{-}$ F4/80$^{+/-}$) (Fig. 3b and Supplementary Fig. 3a). A similar cell progeny was generated in the presence of GM-CSF. However, the MLL-ENL-derived granulocyte population obtained in the presence of GM-CSF was much smaller than in the presence of IL-3, accounting for only 26% of the total cells (Fig. 3b and Supplementary Fig. 3b). Finally, Flt3L stimulation, previously reported to drive dendritic cell (DC) maturation[16,30,31], showed a consistent reduction of Cd11b and Gr-1 expression when compared to Parental-derived cells (Fig. 3b and Supplementary Fig. 3b).

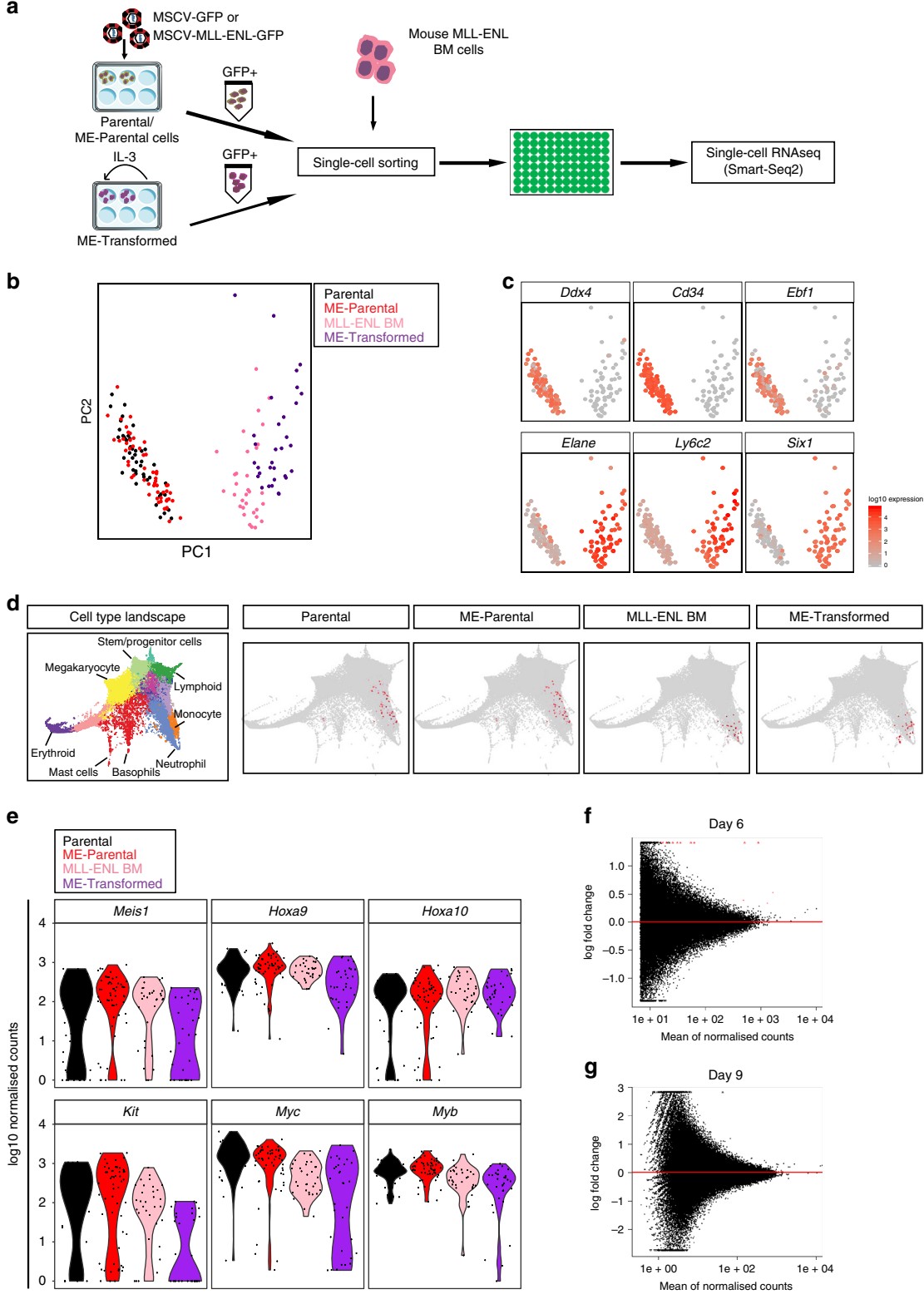

We also assessed cell proliferation during a 7-day time course of myeloid differentiation. For IL-3 and Flt3L differentiation conditions, ME-Parental cells displayed a statistically significant increase in cell numbers compared to the Parental samples (Fig. 3c). This observation is consistent with the phenotypic analysis above given the known reduction in proliferation in mature myeloid cells. Cell cycle promotion by MLL-ENL was also confirmed by flow cytometry, which revealed a trend in the decrease in G1 and an increase in S and G2 phase, evident in the IL-3 and Flt3L treatments (Fig. 3d and Supplementary Fig. 3c). Taken together, induction of myeloid differentiation reveals early cellular consequences of MLL-ENL expression culminating in reduced terminal myeloid differentiation and increased cell proliferation.

**Fig. 2 MLL-ENL fusion gene lacks leukaemic transforming ability in Flt3L and β-estradiol culture condition. a** Outline of experimental strategy. Single cells -Parental and ME-Parental cells cultured in the presence of Flt3L and β-estradiol, ME-Transformed and MLL-ENL BM cells cultured long term in the presence of IL-3- were sorted into 96-well plates and processed for scRNA-seq using Smart-Seq2 protocol. **b** PCA plot based on highly variable genes of 166 cells. Parental (47 cells) and ME-Parental (55 cells) cells are Hoxb8/Flt3L dependent while MLL-ENL BM (32 cells) and ME-Transformed (32 cells) are cultured long term in the presence of IL-3. **c** Expression of selected genes in cells in (**b**). Selected genes are important in defining PC1 component. Top genes are MPP4/LMPP-specific genes; bottom genes represent myeloid and leukaemogenic genes. Cells are coloured according to the expression levels of denoted genes. Colour scheme is based on log10 scale of normalised counts from 0 (grey) to 4 (red). **d** Projection of transcriptomic profiles of Parental, ME-Parental, ME-Transformed and MLL-ENL BM cells onto force-directed graph obtained from single HSPCs. First panel shows the cell type landscape and clustering generated by Dahlin et al.[24]. Following panels show the most similar cells of the landscape to Parental, ME-Parental, ME-Transformed and MLL-ENL BM cells. **e** Violin plots showing expression levels (log10 scale of Normalised counts on y-axis) of key leukaemic target genes across Parental, ME-Parental, MLL-ENL BM and ME-Transformed cells. **f, g** MA plots showing the comparison between counts obtained at Parental and ME-Parental accessible regions defined via ATAC-seq at day 6 (**f**) or day 9 (**g**) of culture in the presence of β-estradiol and Flt3L. No statistical differences can be detected except for regions corresponding to the transduced MLL-ENL (depicted in red).

**Early molecular changes during MLL-ENL transformation.** Having defined early cellular consequences of MLL-ENL expression following the induction of myeloid differentiation, we next explored the corresponding molecular changes. Given the strong phenotype observed after 7 days of differentiation, we performed scRNA-Seq on Parental and ME-Parental cells following 7 days of differentiation in IL-3, GM-CSF and Flt3L together with ME-Transformed and MLL-ENL BM cells (Supplementary Fig. 4a, b). IL-3 was the most effective cytokine to produce cells similar to ME-Transformed and MLL-ENL BM cells. We therefore concentrated our analysis on IL-3-differentiated ME-Parental and Parental cells (Fig. 4a). Cells were first stained with myeloid differentiation surface markers (Fig. 3b and Supplementary Fig. 3b), then GFP + ME-Parental and Parental single cells were sorted and processed using the Smart-seq2 protocol. PCA of the single-cell transcriptomes revealed two MLL-ENL sub-populations (hereafter referred to as MLL-ENL 1 and 2) (Fig. 4b and Supplementary Fig. 4c). Of note, there was a trend towards higher expression of the transgene in MLL-ENL1 cells compared to MLL-ENL2 cells, although it did not reach statistical significance (p value > 0.05), as shown in Fig. 4c. These two populations most likely reflect heterogeneity in the response to the initial change of conditions, as opposed to the more homogenous nature of ME-Transformed cells, which have been cultured in IL-3 long-term (Fig. 2b).

Retrospective analysis of index sort data (Fig. 4d and Supplementary Fig. 4d) showed that the majority of MLL-ENL2 and Parental cells resembled conventional DCs (MHCII+ CD11c+ B220−F4/80+ CD11b+ Gr-1+)[16] and a few resembled macrophages (MHCII+ CD11c+ B220+ F4/80+ CD11b+ Gr-1+)[32,33], while the MLL-ENL1 sub-population was made up of cells displaying a granulocytic phenotype (CD11b^low and Gr-1+). Differential expression analysis confirmed the phenotypic characterisation identified via flow cytometry, since the MLL-ENL1 population showed elevated expression of the neutrophil-related genes Mpo and Prtn3[34–36] with low expression of MHC class II genes such as H2-Ab1 and H2-Eb1, known to be expressed on the surface of DCs and macrophages[37,38] (Fig. 4e).

To define the early molecular changes associated with MLL-ENL expression, pairwise differential expression analysis was performed among MLL-ENL1, MLL-ENL2 and Parental samples (Supplementary Data 2). Genes included for further analysis were selected according to the following parameters: FDR < 0.1 and base mean expression value for each gene larger than 30. Unsupervised hierarchical clustering identified three different gene clusters (Fig. 4f). Genes in cluster 1 (C1), which included mitotic cell cycle genes, were higher expressed in the MLL-ENL1 population. MLL-ENL2 and Parental populations expressed higher levels of genes contained in clusters 2 (C2) and 3 (C3), which included immune response-related genes. Additional GO

categories were identified using the FastProject tool[39] for exploration of gene signatures using two-dimensional projections such as PCA. Figure 4g shows that MLL-ENL1 was particularly enriched for gene sets such as "Formation of translation preinitiation complex", "DNA replication initiation" and "glycine metabolic process"[25,40,41] which may be associated with the increase in cell cycle gene expression previously shown; and "positive regulation of telomerase activity", previously reported as a promising target for AML cell eradication[42].

Following on from bioinformatic analysis of day 7 cells only, we next explored to what extent these early transcriptomic changes might foreshadow the transcriptional events characteristic of fully transformed MLL-ENL cells adapted to grow long-term in IL-3. We therefore repeated the PCA from Fig. 4b (corresponding to Parental and ME-Parental cells differentiated in IL-3 for 7 days), including the single-cell transcriptomes of MLL-ENL BM and ME-Transformed cells (corresponding to fully transformed MLL-ENL cells). As shown in Fig. 4h, only MLL-ENL1 clustered together with MLL-ENL BM and ME-Transformed cells, all expressing the neutrophil marker gene Elane (Fig. 4i). By contrast, MLL-ENL2 and Parental cells clustered separately from all the other samples, and expressed high levels of Cd74, an antigen-presenting cell marker. Known target genes of MLL fusion proteins were elevated in MLL-ENL1 as well as the MLL-ENL BM and ME-Transformed cells, while regulators of myeloid differentiation showed reduced expression (Fig. 4j). To understand the dynamic appearance of the MLL-ENL1 population, we repeated the transcriptomic analysis at days 4, 7 and 11 of differentiation in the presence of IL-3 (Supplementary Fig. 4e). MLL-ENL1 cells can already be distinguished at day 4, but they become more distinct by day 7 and increase in number by day 11.

**Integration with a CRISPR screen identifies candidate targets.** Having identified that early molecular changes foreshadow the MLL-ENL preleukaemic transcriptional program, we next explored whether any of these early events represent genetic vulnerabilities associated with MLL-ENL expression by performing a genome-wide CRISPR-Cas9 drop-out screen in both the Parental (Flt3L and β-estradiol dependent) as well as the ME-Transformed cells cultured in IL-3 (Fig. 5a). Cells were transduced with a genome-wide guide RNA (gRNA) lentiviral supernatant containing 90,230 guides targeting a total of 18,424 mouse genes (average of 3–5 guides per gene)[43]. Cell aliquots were harvested at days 6, 10 and 12 post transduction and gRNA representation determined by next-generation sequencing.

The genome-wide screen, performed using at least two biological replicates per cell line, was analysed using MaGECK[44] and this revealed 465, 1624 and 1798 depleted genes (FDR < 0.25)

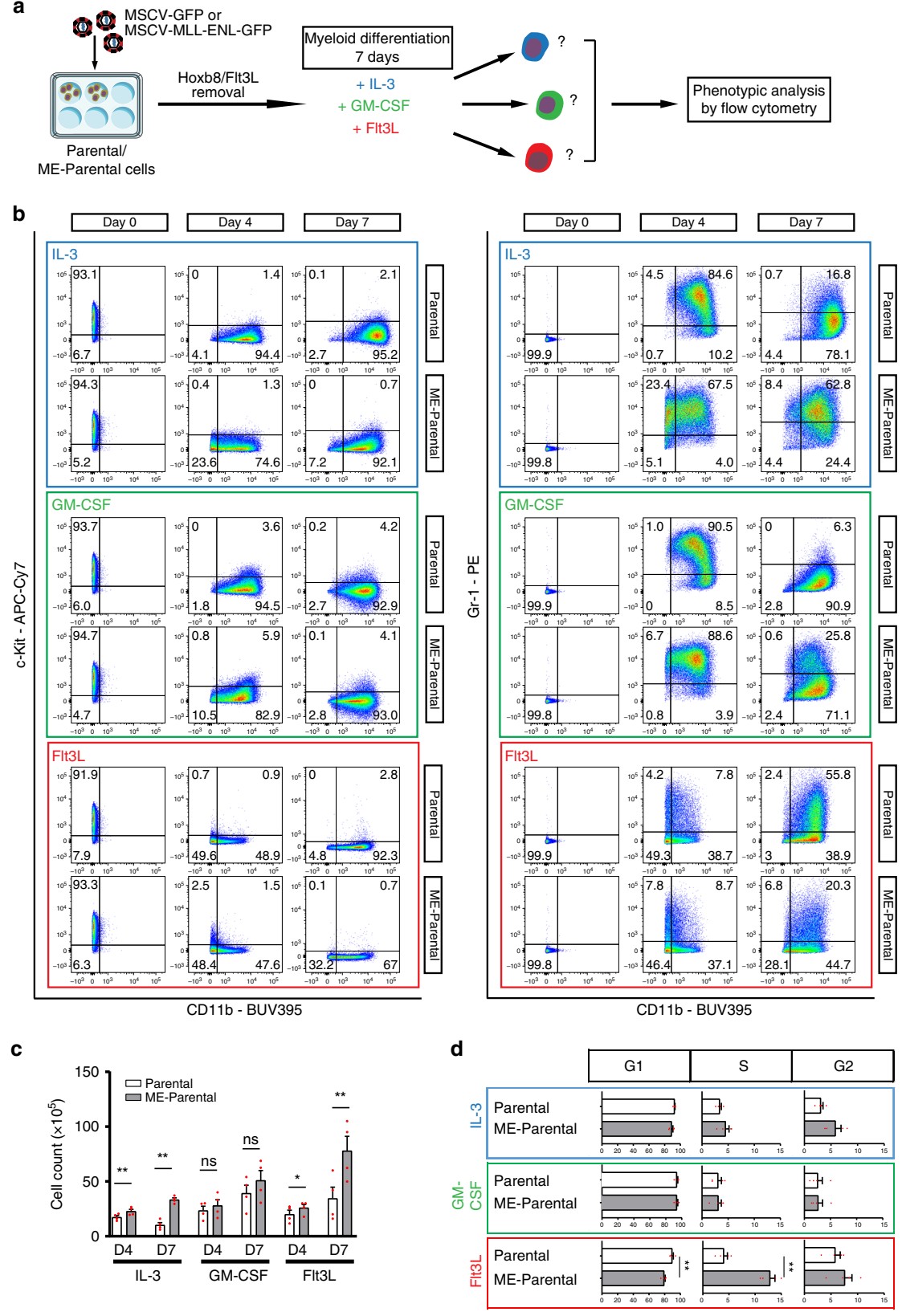

for d6, d10 and d12 ME-Transformed cells respectively, in line with the number of drop-outs obtained using an equivalent gRNA library on multiple AML cell lines[43]. As for the Parental cell line, 1123 and 1440 depleted genes were identified for d6 and

d10 time points respectively (Supplementary Data 3). Parental cells were also collected at day 12 but not further analysed due to an apparent loss of guide complexity. Notably, *Flt3* and *Il3ra*, key essential genes for the survival of the Parental (dependent on

**Fig. 3 The MLL-ENL fusion gene delays CD11b expression. a** Schematic diagram representing the outline of in vitro myeloid differentiation using IL-3, GM-CSF and Flt3L. ME-Parental and Parental cells were obtained by transduction of Hoxb8-FL cells with either MLL-ENL or empty vector control, respectively. After removal of Flt3L and β-estradiol, Parental and ME-Parental cells were differentiated in the presence of either IL-3, GM-CSF or Flt3L. Cultures were then analysed by flow cytometry after 4 and 7 days of differentiation taking the initial culture (day 0) as reference. **b** Phenotypic analysis by flow cytometry of Parental and ME-Parental samples after culturing in presence of either IL-3, GM-CSF or Flt3L. Data were acquired after 4 and 7 days of differentiation. Day 0 represents cells before treatment (in Flt3L and β-estradiol culture condition). Representative plots of three ($N = 3$) biologically independent experiments are shown together with mean values for each gate. **c** Bar charts representing cell counts following 4 and 7 days of differentiation in either IL-3, GM-CSF or Flt3L for Parental and ME-Parental cells. Values are expressed as mean ± SEM. $N = 4$ biologically independent experiments. Statistics were determined by two-tailed paired $t$ test. $P$ values for each comparison (from left to right): 0.01, 0.001, 0.0983, 0.1627, 0.0296 and 0.0075, denoted as *$p <$ 0.05; **$p \leq 0.01$. **d** Cell cycle analysis by flow cytometry of Parental and ME-Parental cells after 7 days of myeloid differentiation. Values are expressed as mean ± SEM. $N = 4$ biologically independent experiments. Statistics were determined by two-tailed paired $t$ test. Only statistically significant differences are labelled; $p$ values are 0.0080 and 0.0033 for Flt3L G1 and Flt3L S, respectively, both denoted as **. Source data are provided as a Source Data file.

Flt3L) and ME-Transformed (dependent on IL-3) cells, were significantly depleted in the respective screens ($p$ values of 2.69E−07 and 8.07E−07 at d6 and d10 respectively for *Flt3* in the Parental cells and 0.1175, 6.32E−05 and 9.55E−05 at d6, d10 and d12 respectively for *Il3ra* in the ME-Transformed cells). Overall, these data confirmed the efficiency of the screen and the biological relevance of drop-out genes.

To identify genetic vulnerabilities specific to the ME-Transformed cells, we merged, for each cell line, drop-out genes from all time points and intersected the resulting gene lists from both cell lines (Fig. 5b, c and Supplementary Data 4). As expected, the 1171 genes that dropped out in both the Parental and ME-Transformed cells showed enrichment for essential biological processes such as "Metabolism of RNA" and "CDK regulation of DNA replication", and 548 specific genes for the Parental cell line showed enrichment for "Metabolism of RNA" and "BRCA1-PCC network"[45,46]. The 897 genes that dropped out specifically in the ME-Transformed cells on the other hand showed enrichment for gene ontology classifications that included "mitotic cell cycle", "chromatin organisation", "ATM pathway" and "Chronic myeloid leukaemia"[47,48].

To focus on MLL-ENL-specific drop-out genes that are associated with early transcriptional changes, we next compared the 897 MLL-ENL-specific drop-outs with the 1553 differentially upregulated genes between the MLL-ENL1 population (defined following differentiation in IL-3) and the Parental cell line. The overlap of 127 shared genes (Fig. 5d and Supplementary Data 5) included genes with potential clinical relevance such as DHODH (currently involved in AML and MDS clinical trials) and PRMT1 (described to play roles in haematological as well as solid cancers and also involved in clinical trials). Overrepresented gene set enrichment analysis (GSEA) categories included "mitotic cell cycle" and "Activation of ATR in response to replication stress" (Fig. 5e). The DDR pathway represents an attractive therapeutic concept in cancer therapy especially in the context of radio- and chemotherapy combinations, as well as synthetic lethal approaches[47]. Moreover, there is pre-clinical evidence that inhibition of DDR mediators, such as ATM and ATR, may represent potential therapeutic strategies for AML[49,50]. Of note, none of these 127 genes were upregulated in Parental cells after 7 days of culture in the presence of IL-3. We therefore interrogated the "druggability" of the 127 overlapping genes using the Drug Gene Interaction Database (DGIdb)[51], which reported 47 (37%) genes to be in druggable categories (Fig. 5f and Supplementary Data 5). Taken together therefore, the genome-wide CRISR-Cas9 screen allowed us to identify a number of genetic vulnerabilities that are associated with early transcriptional changes following transformation and are specific to the ME-Transformed cells.

**Validation of *Atm*, *Cdc7* and *Ldha* as candidate drug targets.** From the 47 druggable genes, inhibitors were readily available for

three genes, allowing us to perform initial validation experiments confirming them as potential therapeutic targets in AML (Fig. 5f). Following confirmation of the genome-wide CRISPR-Cas9 screen by individual gRNA targeting (Supplementary Fig. 5a), we first tested inhibition of the Serine/Threonine Kinase gene (*Atm*), which had been shown before to prolong survival of mice injected with MLL-ENL cells[49]. Inhibition of the Cell Division Cycle 7 gene (*Cdc7*) was another promising target due to its involvement in DNA repair[52] and studies suggesting its importance in both solid[53] and liquid cancers[54]. Lactate Dehydrogenase A gene (*Ldha*) had not been explored in AML but overexpression has been implicated in a range of solid cancers[55,56]. Interestingly, we were able to show that high *Ldha* expression also correlated with poor patient survival in the Leukaemia Gene Atlas (LGA) platform (Supplementary Fig. 5b).

To assess the consequences of small molecule-based target gene inhibition, we used colorimetric assays (MTS assay) to determine the IC50. Effects on the ME-Transformed cells were compared to Parental and ME-Parental cells cultured in self-renewal conditions, to take advantage of our sequential stages of transformation and thus assess potential selectivity in targeting the transformed state. As expected from previous reports[49,57], the ATM inhibitor (ATMi) (AZD0156) consistently caused a marked growth inhibition when measured at both 48 and 72 h (Fig. 6a and Supplementary Fig. 6a). Importantly, the IC50 of 22.9 nM for the ME-Transformed sample at 48 h was sixfold lower compared with the Parental and ME-Parental samples (146.6 and 135 nM respectively). Additionally, this difference increased further after 72 h (14.2-fold decrease of ME-Transformed IC50 value compared to the negative controls), and was highly significant ($p$ value < 0.001 at 48 h and <0.0001 at 72 h).

To test potential effects of the CDC7i (PHA-767491), we exposed cells to higher concentrations (2000, 4000, 5000, 6000, 7000 and 8000 nM) since we did not observe effects up to 1000 nM. As shown in Fig. 6b, ME-Transformed had lower metabolic activity and therefore lower IC50, compared to the negative controls (Supplementary Fig. 6b). This difference became more evident after 72 h of treatment: while ME-Transformed IC50 decreased over time, Parental and ME-Parental showed an opposite trend with an increase of IC50s from 48 to 72 h ($p <$ 0.001 at 48 and 72 h). To conclude, CDC7i had less impact on cell metabolic activity compared to ATMi but showed inhibitory effects with selectivity for the ME-Transformed cells, thus validating the results of the CRISPR-Cas9 screen.

The third target gene *Ldha* selected for validation codes for a metabolic enzyme that is able to convert pyruvate into lactate. Much higher doses of the LDHAi (GSK 2837808A) compound had to be used to observe an effect compared to previously described ATMi and CDC7i treatments, possibly related to the high *Ldha* expression levels compared with *Atm* and *Cdc7* (Fig. 5f). Nevertheless, the MTS assay confirmed selectivity, because LDHAi caused a reduction in the metabolic activity of

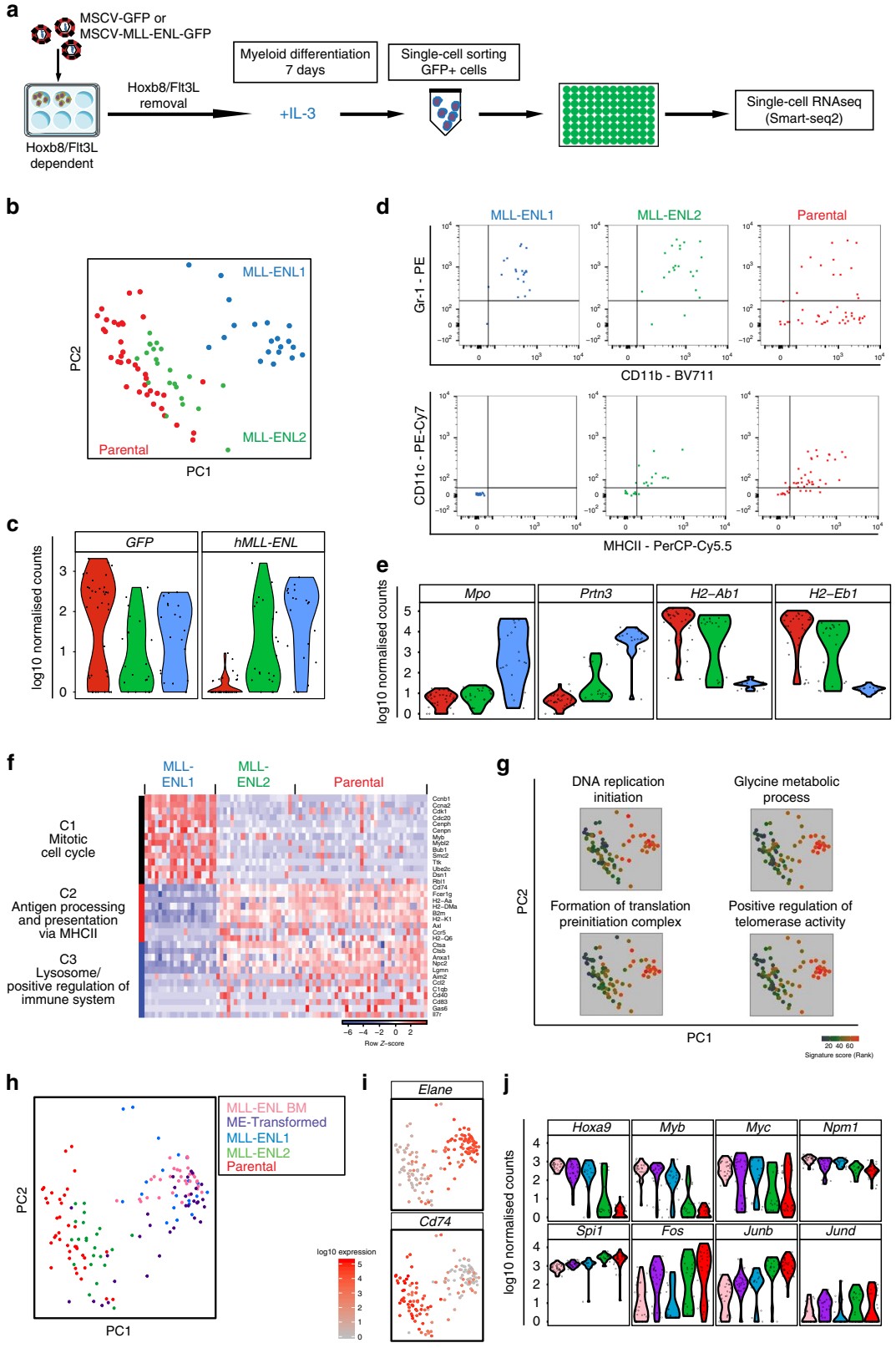

ME-Transformed cells compared to Parental and ME-Parental cells 72 h post treatment, with IC50 values of 83.9, 118.5 and 114.5 μM respectively (ME-Transformed vs. Parental p value < 0.001 and ME-Transformed vs. ME-Parental p value < 0.01) (Fig. 6c and Supplementary Fig. 6c). Overall, these results suggest

that our drop-out screen with matched parental and transformed cells has indeed identified drug targets with a potential therapeutic window.

Finally, another compound targeting the DNA repair gene *Atr* was tested. The ATR pathway was a significantly enriched

**Fig. 4 IL-3 stimulation captures early events of MLL-ENL leukaemic transformation. a** Schematic diagram of in vitro IL-3 scRNAseq. Parental and ME-Parental cells, after Flt3L and β-estradiol removal, were cultured with IL-3 for 7 days, then single GFP + cells were sorted and processed using Smart-Seq2. **b** PCA plot based on highly variable genes of Parental (37 cells) and ME-Parental (42 cells) cells differentiated in IL-3 for 7 days. ME-Parental cells were subdivided into MLL-ENL1 (20 cells) and MLL-ENL2 (22 cells) according to their similarity to Parental differentiated cells. **c** Violin plots showing distribution and expression (log10 scale of normalised counts) of GFP and hMLL-ENL genes across IL-3 differentiated Parental (red), MLL-ENL1 (blue) and MLL-ENL2 (green) samples. **d** Flow cytometry levels of CD11b, Gr-1 and MHCII for cells analysed by scRNA-Seq in panel (**b**) obtained by index sort. **e** Violin plots showing distribution and expression of granulocyte genes (*Mpo* and *Prtn3*) and antigen-presenting cell markers (*H2-Ab1* and *H2-Eb1*) across IL-3 differentiated samples analysed in panel (**b**). **f** Heat map showing the top GO categories defined by GSEA by *Z*-score transformed expression of the most differentially expressed genes specific for each GO category: C1, "Mitotic cell cycle" (adjusted p value = 3.49E-21); C2 "Antigen processing and presentation via MHCII" (adjusted p value = 6.65E-13); C3 "Lysosome/Positive regulation of immune system" (adjusted p values = 7.78E-06 and 2.37E-05, respectively). **g** GO annotation based on the PCA plot using FastProject[39] (significance score of GO terms with red as the highest and green as the lowest). **h** PCA based on highly variable genes including IL-3 differentiated cells (Parental, MLL-ENL1 and MLL-ENL2) and cells cultured long term in IL-3 (ME-Transformed and MLL-ENL BM). **i** Expression of selected genes in cells in panel (**h**). Selected genes (*Elane*, *Cd74*) are important in defining PC1 component. Cells are coloured according to the expression levels of denoted genes (log10 scale of normalised counts from 0 (grey) to 5 (red)). **j** Violin plots showing distribution and expression of known MLL-fusion target genes (top genes) and regulators of myeloid differentiation (bottom genes) across all samples analysed in panel (**h**).

biological process for the 127 genes depleted in the drop-out screen and upregulated at early phases of leukaemogenesis (Fig. 5e). However, the *Atr* gene itself was not significantly depleted in the drop-out screen. Treatment of cells with ATRi (AZD6738) agreed with the results of the screening, showing no specific effect for the MLL-transformed cells. Figure 6d and Supplementary Fig. 6d show the IC50 values obtained after the exposure to ATRi for 48 and 72 h, with no significant difference (p value > 0.05) between ME-Transformed and control samples at any of the concentrations tested. ATRi treatment therefore further validated our drop-out screen. Overall, the small molecule inhibitor analysis demonstrated that our integrated analysis of the CRISPR-Cas9 drop-out screens and scRNA-seq analysis provides a useful platform to identify potential therapeutic targets.

## Discussion

MLL translocation fusion proteins represent some of the most commonly used oncogenes for the generation of leukaemia models generated to date. Nevertheless, key questions about the leukaemogenic mechanisms remain unanswered, and contradictions between individual studies highlight the complex interplay of multiple parameters such as type of MLL-r, microenvironment, oncogene delivery method and cellular context. We took advantage of cells conditionally blocked at the multipotent haematopoietic progenitor stage to develop an MLL-r model with a clear clonal relationship between the parental and MLL-leukaemic cells. Through a combination of scRNA-seq coupled with genome-scale CRISPR-Cas9 screening and inhibitor assays, we highlight genes and pathways likely to be crucial during early leukaemogenic evolution of the disease.

How MLL leukaemia can be initiated from different cell types along the haematopoietic hierarchy is still under debate. MLL-driven leukaemic transformation has been described in HSC, CMP, GMP, CLP, MPP and LMPP[5–7,9,18,19,58]. Of note, MLL-ENL was shown not to induce AML in mice when transduced into highly purified HSCs[9] and require myeloid differentiation for efficient leukaemic transformation[29], contradicting another study suggesting that the HSC compartment is more susceptible to transformation than GMPs[8]. However, the latter study sorted Lin-c-Kit+Sca-1+CD34− cells, which include HSCs as well as MPPs[13]. Importantly, single-cell molecular profiling studies emphasise the notion that all classical populations purified by flow cytometry display substantial heterogeneity[17,59], which means that the exact nature of the target cell for transformation will remain obscure when using these conventionally defined populations.

To overcome these limitations, we took advantage of the conditionally blocked in differentiation and cytokine-dependent mouse haematopoietic progenitor cell line Hoxb8-FL[16] to model MLL-ENL-induced leukaemia, which mirrors the behaviour of classically derived MLL-ENL cell lines both in vitro and in vivo[18,19], but in addition also shows a clear linear relationship between the parental and transformed states. Of note, our AML model revealed that expression of MLL-ENL only had very limited impact on the transcriptome as well as open chromatin status of the Hoxb8-FL cells as long as they were cultured in multipotent, self-renewal conditions (ME-Parental cells), reminiscent of a previous report suggesting that highly purified HSCs are intrinsically protected against MLL-ENL-mediated transformation[9]. However, MLL-ENL-mediated transcriptional dysregulation was readily captured in our model when ME-Parental cells were exposed to an adequate stimulus, for instance myeloid differentiation[29], which allowed us to demonstrate that (i) gene expression changes during early myeloid differentiation correspond to immediate activation of the leukaemogenic program, and (ii) some of the transduced cells differentiated normally despite the expression of MLL-ENL, thus enhancing our broader understanding of the cellular permissiveness for AML development.

Our full transcriptome analysis of expression changes associated with early transformation was perfectly suited to being coupled with genome-scale CRISPR drop-out screens to prioritise genes and pathways based on their selective importance for transformed cell growth. This included the DDR (DNA Damage Response) and several metabolic pathways, some of which had already been described in MLL-r frank leukaemia[41,49,60], but our MLL-ENL model was able to identify their importance also at early phases of transformation. It is important to note that the Parental cells grow faster than the ME-Transformed cells, yet the DNA repair and cell cycle genes identified in here were specific drop-outs in the ME-Transformed cells. These genes were also upregulated in the expression analysis. This can be interpreted as an indication that ME-Transformed cells have adopted a cellular state where cell division (proliferation) is counterbalanced by pushes to differentiate and/or die. Upregulation of genes such as ATM may be required to mitigate the resulting strain, consistent with our observation that even though the ME-Transformed cells express higher levels of these genes, they are nevertheless more sensitive to the inhibitors. Small molecule inhibitors to CDC7 and LDHA showed selective activity in the transformed cells compared with their parental counterparts, although the

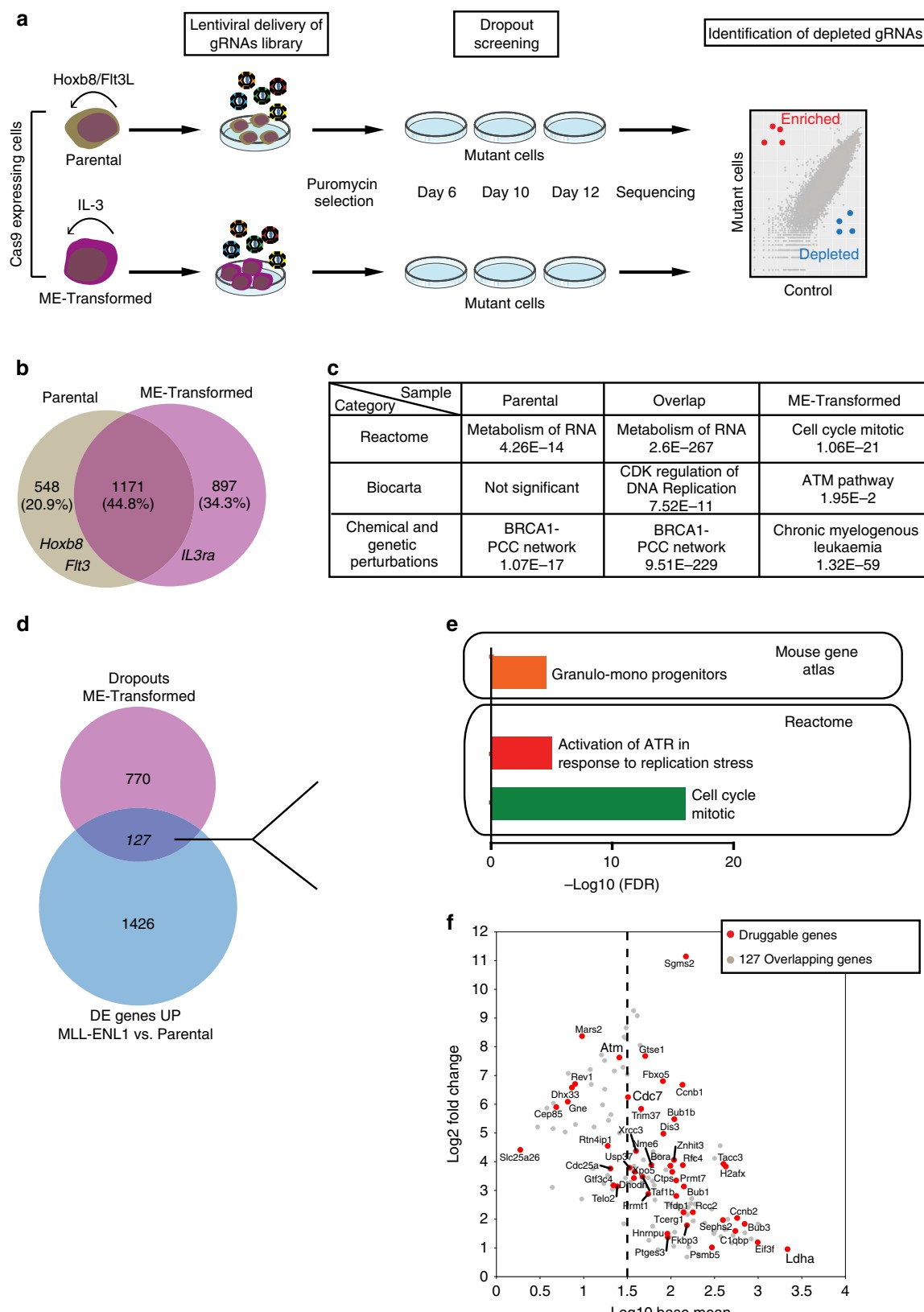

concentrations of compound required were higher than for the ATM protein, especially in the case of LDHA and in line with previous reports using solid cancer cell lines[61,62]. Future development of more potent inhibitors may unlock the targeting of metabolic pathways as viable treatment strategies in AML.

## Methods

Materials and methods are summarised below, with more detailed experimental protocols provided in Supplemental Information.

**Retroviral production and transduction of Hoxb8-FL cell line.** pMSCV-neo retroviral vector, containing human MLL-ENL (hMLL-ENL) cDNA[63], was digested

**Fig. 5 Genome-wide drop-out screens identify new therapeutic targets. a** Summary of the experimental approach. Parental and ME-Transformed cells were obtained as described in Fig. 1a from Cas9-expressing Hoxb8-FL cells. Then, Parental cells (cultured in self-renewal conditions) and ME-Transformed cells (IL-3 dependent) were infected with a lentiviral pool bearing a genome-wide gRNA library. Forty-eight hours post transduction, puromycin was added in order to select for efficiently infected cells. Cell aliquots were then sampled at days 6, 10 and 12 post infection. DNA was extracted, gRNAs were PCR amplified and subsequent libraries were sequenced. Enriched or depleted gRNAs were determined by comparison to the library used for infection. Depleted guides represent genes whose expression is required for cell survival. **b** Intersection of drop-outs selected with FDR < 0.25 of Parental and ME-Transformed samples. **c** Table showing top significant pathways enriched in each cell subgroup identified in (**b**). The GSEA database was used and an FDR < 0.05 was applied to define a pathway as statistically significant. Individual FDR values are shown. **d** Venn diagram showing intersection between unique ME-Transformed drop-outs and upregulated genes obtained from comparing the transcriptomic profile of MLL-ENL1 and Parental cells (refers to Fig. 4). **e** GSEA analysis of pathways using the 127 overlapping genes identified. An FDR < 0.05 was applied to define a pathway as statistically significant. **f** MA plot showing expression levels of each of the 127 overlapping genes in ME-Transformed cells (expressed as log10) against its difference of expression in the comparison MLL-ENL1 vs. Parental (expressed as log2 of the fold-change). Genes defined as druggable are highlighted in red; genes selected for validation (Supplementary Fig. 5) and further analysis are in bold.

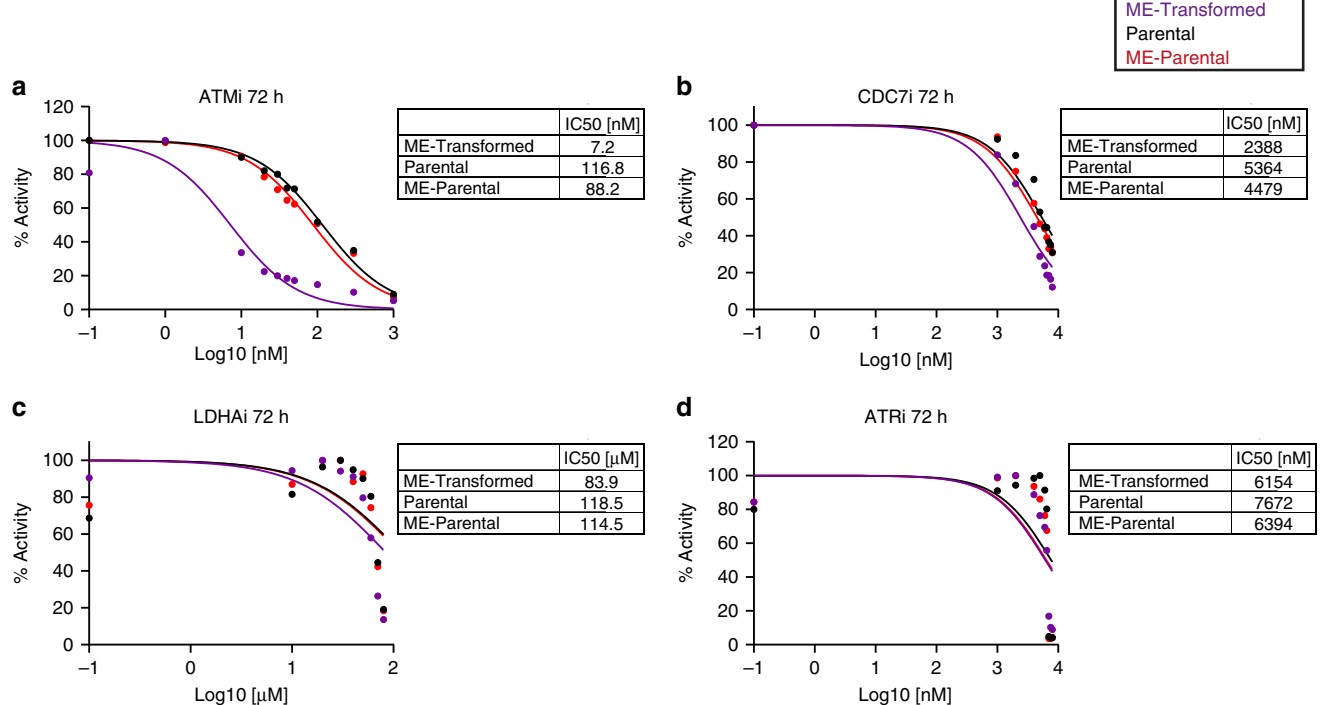

**Fig. 6 ATMi, CDC7i, LDHAi and ATRi treatments validate the genome-wide CRISPR screen. a–d** Plots showing IC50 fitting curves and IC50 values at 72 h post treatment of Parental, ME-Parental (cultured in presence of Flt3L and β-estradiol) and ME-Transformed (cultured in the presence of IL-3) with inhibitors against ATM (**a**), CDC7 (**b**), LDHA (**c**) and ATR (**d**). Data are shown as mean of biological replicates; N = 2 biologically independent experiments. Source data are provided as a Source Data file.

in order to remove neo resistance. IRES-eGFP sequence was PCR amplified from pMSCV-PIG-IRES-eGFP vector (Addgene) using KAPA HiFi HotStart ReadyMix PCR Kit (KAPA BIOSYSTEMS) and cloned into MSCV-MLLENL. Retroviral production and transduction of Hoxb8-FL cells was performed via spinoculation as indicated in Supplementary Methods, also described in ref. [64]. Parental and ME-Parental GFP + cells were FACS sorted 6 days after spinoculation using a Melody cell sorter (BD Biosciences). Sorted cells were then cultured typically for 3 more days before performing further experiments.

**Cell culture**. Parental cells (Hoxb8-FL cells) were cultured in RPMI 1640 medium (Sigma-R8758) supplemented with 10% fetal bovine serum (FBS) (HyClone™ GE Healthcare), 1% penicillin/streptomycin (Sigma-P0781), 1% L-Glutamine (Sigma-G7513), 0.1% 2-Mercaptoethanol (50 mM stock)(Gibco®), cell culture supernatant from an Flt3L-producing B16 melanoma cell line (5% final concentration) and 1 μM β-estradiol (Sigma-E2758), as originally described[16]. ME-Parental cell line was cultured as Parental cells. ME-Transformed and MLL-ENL BM were cultured in RPMI 1640 medium (Sigma-R8758) supplemented with 10% FBS (HyClone™ GE Healthcare), 1% penicillin/streptomycin (Sigma-P0781), 1% L-Glutamine (Sigma-G7513), 0.1% 2-Mercaptoethanol (50 mM stock) (Gibco®) and 10 ng/ml

of recombinant murine interleukin 3 (IL-3) (PeproTech). All cell lines were kept at a concentration of 1–10 × 10^5 cells/ml and the medium was replenished every 1–2 days.

**Generation of MLL-ENL (ME)-Transformed cell line**. Hoxb8-FL cells were transduced with pMSCV-MLL-ENL-IRES-eGFP retroviral vector. ME-Transformed cell line was then generated by serial re-plating on methylcellulose[5]. Briefly, GFP-positive cells were FACS sorted, washed twice and seeded on methylcellulose as for colony-forming assays (see below). Colonies were recovered, washed and re-plated every 7 days. At the third re-plating, colonies were recovered, washed and transferred to liquid culture containing RPMI 1640 medium (Sigma-R8758) supplemented with 10% FBS (HyClone™ GE Healthcare), 1% penicillin/streptomycin (Sigma-P0781), 1% L-Glutamine (Sigma-G7513), 0.1% 2-Mercaptoethanol (50 mM stock) (Gibco®), 10 ng/ml of recombinant murine interleukin 3 (IL-3) (PeproTech), 10 ng/ml of recombinant murine interleukin 6 (IL-6) (PeproTech) and 50 ng/ml of recombinant murine stem cell factor (SCF) (PeproTech). After two passages, SCF was removed from the media and following two more passages, IL-6 was also removed. Finally ME-Transformed cells were cultured long term in the presence of IL-3 only.

**Haematopoietic colony-forming assay**. Two hundred GFP-positive MLL-ENL or empty vector-transduced Hoxb8-FL cells (ME-Parental and Parental cells respectively) were FACS sorted in RPMI 1640 with 10% FBS and 1% P/S. Cells were then centrifuged at $300 \times g$ for 5 min, resuspended in 100 μl of RPMI 1640 with 10% FBS and 1% P/S and added to 1.1 ml of M3434 Methocult (Stem Cell Technologies). 1.2 ml of methylcellulose-cell mix was plated in 35 mm dishes in triplicate. Cells were cultured at 37 °C with 5% CO₂. Colonies were counted after 7 days, dissociated in 5 ml of 1× PBS, centrifuged at $300 \times g$ for 5 min, re-suspended in 1 ml of RPMI 1640 with 10% FBS and 1% P/S and counted. For subsequent re-plating experiments, 800 cells were re-plated in 1.1 ml of methylcellulose as described above.

**Myeloid differentiation assay**. ME-Parental and Parental cells maintained in the presence of Flt3L and β-estradiol were washed twice in 1× PBS and $1 \times 10^5$ cells were plated in a six-well plate in 1 ml of myeloid differentiation media. Differentiation media consisted of RPMI 1640 medium (Sigma-R8758) supplemented with 10% FBS (HyClone™ GE Healthcare), 1% penicillin/streptomycin (Sigma-P0781), 1% L-Glutamine (Sigma-G7513), 0.1% 2-Mercaptoethanol (50 mM stock) (Gibco®) and either 5% Flt3L conditional media, or 7 ng/ml mGM-CSF (Pepro-Tech) or 10 ng/ml mIL-3 (PeproTech). Cells were cultured at 37 °C and 5% CO₂. Cells were kept in differentiation media for 7 days, and both suspension and adherent cells were counted at days 4 and 7 and diluted to $5 \times 10^5$ cells/ml if necessary.

**Flow cytometry and sorting strategies**. Cells were centrifuged at $300 \times g$ for 5 min, washed twice in 1× PBS and incubated in 50 μl Fc-block (BioLegend) at room temperature for 5 min. Following blocking step, 50 μl of antibody mixture diluted in FACS buffer (1× PBS plus 2%FBS) was added and samples were incubated for 30 min at 4 °C. In parallel, single staining controls using UltraComp eBeads™ Compensation Beads (Thermo Fisher) and Fluorescence Minus One (FMO) were prepared.

Single-cell sorting for Smart-Seq2 was performed using an Influx cell sorter (BD Biosciences, San Jose, CA). Cells were sorted into lysis buffer and processed as described below. The LSRFortessa (BD Biosciences) was used to analyse the cells. The flow cytometry data were analysed using FlowJo software v10.6.1 (Treestar, Ashland, OR).

Antibodies used are listed in Supplementary Methods.

**Cell cycle analysis**. Cells were centrifuged at $300 \times g$ for 5 min and stained in 500 μl of 20 μg/ml Hoechst 33342 (Biolegend) for 45 min at 37 °C. Cells were then centrifuged at $300 \times g$ for 5 min at 4 °C, washed in cold medium and resuspended in 500 μl of cold medium in addition with 7-aminoactinomycin D (Thermo Fisher Scientific) (1:125). The LSRFortessa (BD Biosciences) was used to run and analyse the cells.

**Single-cell RNA sequencing (scRNA-seq)**. Cells were single-cell sorted by FACS directly into individual wells of a 96-well plate containing lysis buffer and processed using Smart-Seq2 protocol[20]. Libraries were prepared from ~150 pg of DNA using the Illumina Nextera XT DNA preparation kit. Pooled libraries were run on the Illumina HiSeq 4000, then raw reads were aligned to *Mus musculus* genome (GRCm38.81) using G-SNAP (version 2015-09-29) with the following parameters: –B 5 –n 1 –N 1 –Q. HTSeq-count (version 0.6.0) was run to assign mapped reads to Ensembl genes (GRCm38.81).

**ATAC-seq**. Three pools of 50,000 ME-Parental and Parental cells each, cultured in the presence of Flt3L and β-estradiol, were bulk sorted into 1.5 ml tubes and processed following the established ATAC-seq protocol[28]. Samples were sequenced on the Illumina HiSeq 4000. Reads were aligned to *Mus musculus* genome (GRCm38.81) using Bowtie2 (v2.2.5), obtaining 55% of unique mappable reads. Peak calling was run using F-Seq[65] (v3) with the following parameters: -t14, -f1. Peaks called for either Parental or ME-Parental were considered for further analysis. MA plots for comparison between Parental and ME-Parental were obtained using DESeq2[66] (v1.26.0).

**In vivo injection of ME-Transformed cell line**. For tumour induction, $5 \times 10^5$ ME-Transformed or Parental cells (CD45.1+), together with $2 \times 10^5$ CD45.2+ total bone marrow cells were injected via tail-vein injection into lethally irradiated C57BL/6J mice. Ten mice were used in total: five mice injected with ME-Transformed cells and five mice with Parental cells. Blood samples were taken on days 7, 14, 21 and 31 post transplantation and chimaerism levels were assessed via flow cytometry using GFP, CD45.1+ (Biolegend, clone A20) and CD45.2+ (Biolegend, clone 104) staining.

**Mice**. Six-week-old female C57BL/6J mice were bred and maintained at the University of Cambridge in microisolator cages and provided continuously with sterile food, water, and bedding. All mice were kept in specified pathogen-free conditions, and all procedures were performed according to the United Kingdom Home Office regulations.

**CRISPR-Cas9 screening**. CRISPR-Cas9 genome-wide screening was performed following the methodology described by Tzelepis et al.[43]. Briefly, Cas-9 expressing cells were generated first by lentiviral transduction using pKLV2-EF1aBsd2ACas9-W and blasticidin (10 μg/ml) selection applied 2 days post infection. In order to perform CRISPR-Cas9 screening, Cas-9-expressing cells were infected with genome-wide gRNA lentiviral supernatant. Forty-eight hours post transduction, cells were selected with puromycin. Selected cells were harvested at days 6, 10 and 12 post infection, DNA was extracted and gRNAs libraries were generated for Illumina sequencing (HiSeq2500).

**Reporting summary**. Further information on research design is available in the Nature Research Reporting Summary linked to this article.

## Data availability

The genome-wide CRISPR screening and exome sequencing data referenced during the study have been deposited in the European Nucleotide Archive (http://www.ebi.ac.uk/ena) with numbers ERP118720 and ERS529672 and ERP117027. scRNA-seq and ATAC-Seq data have been deposited in the GEO database (https://www.ncbi.nlm.nih.gov/geo/) under the accession numbers GSE140807 and GSE141353, respectively. Normal bone marrow HSPC dataset published by Dahlin et al.[24] was obtained from GEO database, accession number GSE107727. The source data underlying Figs. 1b–g, 3b–d, 6a–d and Supplementary Figs. 3a–c, 5a, 6a–d are provided as a Source Data file. All the other data supporting the finding of this study are available within the article and its supplementary information files and from the corresponding authors upon reasonable request. A reporting summary for this article is available as a Supplementary Information file.

## Code availability

The code has been deposited in GitHub (https://github.com/SharonWang/Basilico_NCpaper_Code).

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

## Acknowledgements

We are grateful to Reiner Schulte and his team for help with flow cytometry and Dean Pask for help with the mouse transplant assays. S.B. was supported by a CRUK Ph.D. studentship administered through the CRUK Cambridge Centre. Work in the Gottgens group is supported by Wellcome, CRUK, Bloodwise, MRC, NIH-NIDDK and core support grants by the Wellcome and MRC to the Wellcome & MRC Cambridge Stem Cell Institute.

## Author contributions

S.B. designed and performed experiments and wrote the manuscript; A.K. designed and performed experiments and edited the manuscript; K.T., G.G., S.J.K. carried out experiments; X.W. performed bioinformatic analysis and edited the manuscript; P.M.Q. and K.W. performed bioinformatic analysis; D.J.A., L.S.C., B.J.P.H. and G.S.V. supervised parts of the study; F.J.C.-N. designed, supervised and performed experiments and wrote the manuscript; B.G. designed and supervised experiments and wrote the manuscript.

## Competing interests

The authors declare no competing interests. L.S.C. is an AstraZeneca employee (no competing interest).
