## [Peer Review File · Nature Communications]

Reviewers' comments:

Reviewer #1 (Expertise: ATAC, single cell sequencing, Remarks to the Author):

In Basilico S et al., the authors create a new model of MLL-ENL driven acute myeloid leukemia starting from the conditionally immortalized stem cell line Hoxb8-FL. The authors then use this model to ask fundamental questions of leukaemogenesis: What are the early events driving leukemia? Are these early events attractive as potential therapeutic targets? To answer these questions, the authors embark on a multi-omic approach including bulk ATAC-seq and single-cell RNA-seq to understand their model system. Finally, the authors use a genome-wide CRISPR-Cas9 screen to identify potential therapeutic targets. The manuscript is ambitious. However, the relatively limited analysis of their multi-omic data and the translational relevance of the model are the most problematic portions of this paper. We have a significant number of concerns described below.

1. The authors claim to have developed a novel model for AML, however, they do not show any immunohistochemical evidence or flow immunophenotypic evidence of leukemia in mice. Hepatosplenomegaly and decreased survival appear to be the only evidence of "leukemia" in mice. Bone marrow aspirate morphology along with peripheral blood counts would be valuable additional data in support of the development of leukemia.
2. The authors claim a "clear clonal relationship between the parental and leukaemic cells" in their novel model system. Given what we know about clonal heterogeneity, performing whole exome or whole genome sequencing on the ME-parental, preleukemia and BM samples and showing the acquisition of known driver mutations would be powerful additional data.
3. The authors attempt to investigate the transcriptional profile of their new model of early leukemia using single-cell RNA-seq, specifically using a Smart-Seq2 protocol that resulted in profiling 157 single cell transcriptomes. Additional supplemental information regarding the analysis of the scRNA-seq data are necessary. For example, what is the total number of transcripts per cell detected? Basic QC is required.
4. There appears to be significant variability in PC-space (Figure 2Bi) of the BM and ME-Preleukaemia populations. It would be interesting if the authors could comment on the single-cell variability in the RNA-seq data set. Does the slight separation between the ME-Preleukaemia population and the BM population reflect the biology? Or is this an artifact from the experimental system? How many batches of each cell were analyzed (it is well known in the field that batch effects can confound this sort of analysis). Furthermore, it is difficult to interpret the PC analysis without plotting the amount of variance associated with each of the PCs.
5. Figure 2C demonstrates projection of leukemic samples onto normal hematopoietic development. Additional details on how this projection was generated would be very helpful. Identifying all the colored populations explicitly would be very helpful (i.e., what is the pale lavender population? What

is the central red population?). Furthermore, the displays showing the projections of the parental and ME-parental are strikingly similar. It would seem that the probability that both populations would have many cells that project into basically the identical locations on this space is very low. Can the author comment on this significant concern, i.e. what is the probability that both populations what have scattered cells that project to almost identical positions in this space? Can the authors rule out some sort of bug or sample swap? If not, what is the biology underlying this observation?

6. It is unclear why the authors decided to perform bulk ATAC-seq on just the parental and ME-parental cell lines. What was the experimental rationale for thinking that chromatin accessibility would be different between the two populations? Is there evidence in the literature to suggest this? If the authors postulated that chromatin accessibility may play a role in early leukaemagenesis, then why was ATAC-seq not performed on the ME-preleukaemia and BM populations? Finally, given that single-cell RNA-seq was performed on the model system, why did the authors choose to perform bulk ATAC-seq instead of single-cell ATAC-seq on the populations of interest?

7. The ATAC-seq analysis is rather superficial, and thus it would be helpful to see a more detailed investigation/explanation into the ATAC-seq data (Figure 2Ei and 2Eii, Figure S2). The authors demonstrate approximately 10% unique peaks between the two populations – what are enriched in the two distinct peak sets? Are there any transcription factor motifs that are enriched in each population? It would likely be more helpful to see a MA plot rather than the plot shown in figure Eii.

8. Can the authors comment on why the ME-preleukaemia population characterized in Figure 2 does not display the phenotypic heterogeneity observed after only 7 days exposure to IL-3 in figure 4.

9. In Figure 4E, the authors show that MLL-ENL2 clusters with the parental, while MLL-ENL1 clusters with the ME-preleukaemia. How robust is this phenotypic divergence in the MLL-ENL populations? If this experiment was repeated, would you expect to see it again? When do these distinct populations emerge (3 days? 7 days?)? Do they persist? What is driving the divergence of the populations? Are they clonally related?

10. It is not entirely surprising that a drop-out screen in a rapidly proliferating cell line would result in hits to DNA-repair and cell cycle genes. Were there any hits that were surprising (i.e., not involved in DNA repair, cell cycle, glycolytic metabolism)? These might prove to be a more attractive target for potential therapeutics.

11. A figure demonstrating the overall model would be helpful to reference in the discussion.

Reviewer #2 (Expertise: Crispr screen, Remarks to the Author):

In this manuscript, Basilico et al. utilized Hoxb8-FL cells to study the early phase of leukemogenesis driven by MLL-ENL. They compared cells transduced with MLL-ENL retrovirus or a vector control and

found that these two groups showed highly similar profiles in transcriptome and ATAC-seq when cultured under the self-renewal LMPP condition, but MLL-ENL expressing Hoxb8-FL cells exhibited differentiation defects when induced by cytokines. A fraction of IL3-treated Hoxb8-FL cells resembled preleukemic cells by scRNA-seq and retained a higher proliferative capacity. In addition, the authors performed genome-wide CRISPR screens in Hoxb8-FL cells and MLL-ENL-transduced preleukemic cells and found that a subset of genes (~10%) upregulated in pre-leukemic cells is specifically required for pre-leukemic cell proliferation. Three of such genes were inhibited by a small compound and showed expected response.

The experimental system using Hoxb8-FL allows for more accurate characterization of molecular events during the early phase of leukemogenesis, which have been difficult to analyze because of sample availability and cell heterogeneity. The authors applied the scRNA-seq technology to further characterize cells undergoing leukemogenesis. It showed basic characterization of cell populations under investigation; however, apart from the development of the MLL-ENL-based transformation system of Hoxb8-FL cells, it is still not clear what are the key events or molecular mechanisms underlying leukemogenesis. Therefore, this reviewer does not feel that the current version of the manuscript warrants publication in Nature Communications. It would rather be suited in a more specialized journal. Below are the additional major comments for this manuscript:

1. The authors supplied a list of CRISPR screen hits (dropout genes) but no other essential datasets (raw gRNA counts for each cell line, each replicate and each time point, and statistical outputs). Based on the number of dropout genes, the screens seemed to be conducted successfully; however, due to the lack of the dataset, I cannot assess these screens further. All datasets are valuable resource for the community as well. The authors must make the full set of the datasets available.

2. As validation of the screen hits, the authors performed pharmacological inhibition of 3 selected genes. These must be first performed by a genetic means (ie. by gRNA). CDC7i and LDHAI showed very high IC50 and very small difference on sensitivity between preleukemic and parental cells; therefore, it does not support the claim that these genes are specifically required in pre-leukemic cells.

3. Fig. 2C. The plots for Parental and ME-Parental are extremely similar (but does not look completely identical). Are these shown with appropriate datasets? Based on the 'Parental' plot, it seems to me that this population is quite diverged and shows considerable heterogeneity. It would be useful to analyze the scRNA-seq data from the Parental cells more deeply to understand the heterogeneity that exists in Hoxb8 cells and discuss any detrimental effects/consequences for the system developed here.

Minor comment:

Cell labeling is not accurate enough to specify which samples were used. For instance, "Parental" is used as a control, but is this always MSCV-GFP transduced? How long was ME-Parental cell line cultured before each analysis? Is this cell line stable and does not show any spontaneous transformation?

Reviewer #3 (Expertise: Hematopoiesis and AML models, Remarks to the Author):

This is an interesting manuscript that aims to identify the molecular events that precipitate the transition from the preleukemic state to overt leukemia using a novel cell line model expressing MLL-ENL. The authors used the LMPP-like cell line Hoxb8-FL, which is immortalized, but is growth factor-dependent, and can differentiate in response to myeloid growth factors. They transduced these cells with a retrovirus encoding the leukemogenic translocation MLL-ENL, and then cultured these cells in different growth factor conditions to generate what they call "Parental MLL-ENL cells", which are distinct from the "Parental" Hoxb8-FL cells, and "pre-leukemic MLL-ENL cells", which they claim is a model for the clonal succession that is observed in AML patients. The authors show that the "pre-leukemic" cells, but not the parental cells can cause leukemia in mice. They also demonstrate that the pre-leukemic MLL-ENL cells differentiate at a slower rate in response to myeloid growth factors. They then perform a series of single cell RNA sequencing experiments to probe the molecular events that distinguish the 3 different cell types, in an effort to identify the key molecular events associated with an early transition to the leukemic state. Finally, they perform a CRISPR dropout screen MLL-ENL preleukemic and parental cells to identify potential therapeutic targets to prevent leukemic transformation. They validate three of these targets in the same cell lines with pharmacologic inhibitors.

Some strengths of the study are (1) the development of a cell line system that could in theory be used to study clonal succession in AML driven by a variety of mutations or gene rearrangements, and that is amenable to high-throughput screening technologies and (2) the identification of some novel potential therapeutic targets that could be further tested in MLL-rearranged AML. However, the study also has some conceptual flaws and some technical problems.

Major:

1. Unlike some AML mutations and chromosomal rearrangements, MLL-ENL AML has not been demonstrated to have a pre-leukemic phase in patients. Therefore, it is unclear why the authors are calling these "pre-leukemic" cells, and why they chose MLL-ENL to model the pre-leukemic state. Therefore, the clinical significance of this model system is unclear.

2. The parental Hoxb8-FL cell line that is used as the negative control cannot be compared to healthy hematopoietic cells because it is immortalized with Hoxb8. Due to this baseline immortalization, even the parental cells transduced with empty vector could be thought of as “pre-leukemic”, and are in no way normal.

3. In Figure 1, the authors claim that the “ME-pre-leukemic” cells cause leukemia when transplanted into mice, but not the “parental” cells. However, only survival curves and organ weights are shown. To make the diagnosis of AML in these mice, it would be necessary to include histology images of blood, or of bone marrow, spleen and/or liver sections. In addition, flow cytometry analysis of the leukemic cells from the mice should be shown, because these cells could change their cell surface markers in vivo.

4. In Figure 1, transplantation of the “ME-parental” cells is not demonstrated, and it is not clear whether these cells are also capable of causing leukemia. This is critical to the interpretation of the sc RNAseq studies presented later, in which the ME-parental cells are compared with “ME-preleukemic” cells.

Minor:

1. The generation of “ME-parental” cells needs to be better defined. It is not clear from the results section how these cells were generated, and how they are different from “ME-preleukemic” cells.

2. In Figure 3, only representative flow plots are shown, but there is no statistical analysis included, and N=2. This experiment should be repeated at least 3 times, and statistics need to be shown for all of the populations in question.

3. In the table in Figure 3D, no confidence intervals or p values are shown.

4. In the sc RNAseq analysis shown in Figure 4, it is interesting that the MLL-ENL cells were separated into two populations, MLL-ENL1 and MLL-ENL2. However, the biological significance of these two populations is unclear. It would be informative to sort and transplant each of these populations into mice, and determine which one is more efficient at initiating leukemia in vivo.

5. While it is encouraging that some of the candidates from the CRISPR dropout screen could be pharmacologically validated, only the ATM inhibitor appears to be active in these cells at doses that

may be achievable in patients. If this drug could be validated in a mouse model of MLL-rearranged AML, this would increase the clinical relevance of the study.

Reviewer#1: (Expertise: ATAC, single cell sequencing, Remarks to the Author)

This reviewer acknowledges that our approach is ambitious and highlights our multi-omic analysis and the CRISPR-Cas9 genome-wide screen. The reviewer also raises a number of concerns, which we have addressed as outlined below:

1. The authors claim to have developed a novel model for AML, however, they do not show any immunohistochemical evidence or flow immunophenotypic evidence of leukemia in mice. Hepatosplenomegaly and decreased survival appear to be the only evidence of "leukemia" in mice. Bone marrow aspirate morphology along with peripheral blood counts would be valuable additional data in support of the development of leukemia.

We agree that these are all very good points in demonstrating the validity of a leukaemia mouse model. We have therefore now performed histological analysis of organs from leukaemic mice (see Figure S1C).

2. The authors claim a "clear clonal relationship between the parental and leukaemic cells" in their novel model system. Given what we know about clonal heterogeneity, performing whole exome or whole genome sequencing on the ME-parental, preleukemia and BM samples and showing the acquisition of known driver mutations would be powerful additional data.

We believe that this reviewer's comment seems to be the result of us not explaining sufficiently some of the key aspects of our model. Some of this is already stated in our response to the editor's point 1. The Hoxb8-FL cells have a conditional differentiation block, mediated by inducible Hoxb8 expression and requiring supplementation with the cytokine Flt3 ligand. This allows the maintenance of the cells as a self-renewing LMPP pool. Importantly, the cells can be readily grown up from single cells, and therefore clonal populations can be generated. This is why we have a clonal relationship between the starting parental cells, and the leukaemia cells. In a standard retroviral transplantation experiment, many different progenitors are transduced, so that the leukaemia develops from a pool of clones, with distinct target cells of transformation. It is widely accepted that the nature of target cell for transformation is important, but in traditional models, this key aspect remains a bit of a black box, because even when FACS purified populations are used for retroviral transduction, we now know that all of these FACS populations are composed of heterogeneous cells. Our model circumvents this, because we know the exact nature of the target cell for transduction, and then as a consequence have a direct clonal relationship between that well-defined cell and the resulting leukaemia, thus permitting true "before/after" comparisons.

With all that being said, we nevertheless decided to "go the extra mile" and teamed up with David Adams' group (Sanger Institute) to carry out the exome sequencing analysis suggested by the reviewer. Not unsurprisingly for an MLL translocation model, our results show that there was no acquisition of additional known driver mutations during the process. These new results are described now in the manuscript.

3. The authors attempt to investigate the transcriptional profile of their new model of early leukemia using single-cell RNA-seq, specifically using a Smart-Seq2 protocol that resulted in profiling 157 single cell transcriptomes. Additional supplemental information regarding the analysis of the scRNA-seq data are necessary. For example, what is the total number of transcripts per cell detected? Basic QC is required.

We totally agree with the reviewer that some readers may be interested in seeing this type of information, not only as reassurance about the data in the paper, but also if they are interested in downloading the data to use them for their own work. We were the first group to report scRNA-Seq for highly purified HSCs (Wilson et al, CELL STEM CELL 2015), and have since published many papers with this technology. We therefore have a very robust processing pipeline, with extensive QC analysis. The revised version of our study includes now the

transcriptional profile of 1000 cells, with a median of 1,380,562 reads associated to nuclear genes per cell, a median of 6,539 total genes detected per cell, and a median of 4,498 genes detected at levels of at least 10 reads per million. For the revision, we have explained in more detail the QC steps and included this relevant information (see Extended methods).

4. There appears to be significant variability in PC-space (Figure 2Bi) of the BM and ME-Preleukaemia populations. It would be interesting if the authors could comment on the single-cell variability in the RNA-seq data set. Does the slight separation between the ME-Preleukaemia population and the BM population reflect the biology? Or is this an artifact from the experimental system? How many batches of each cell were analyzed (it is well known in the field that batch effects can confound this sort of analysis). Furthermore, it is difficult to interpret the PC analysis without plotting the amount of variance associated with each of the PCs.

We agree that these are interesting questions, because the reviewer is entirely correct when stating that a key issue with single cell analysis is to design experiments in such a way that batch effects can be differentiated from biological effects. We now realize that we should have explained our experimental design much better. As we have done scRNA-Seq for a long time and are acutely aware of the exact issues that the reviewer mentions, we tried to minimise the use of cells from different batches in a single figure. In those cases where cells from several batches needed to be included in a single figure, we always processed one of the populations in the 2 batches so that batch effect could be accounted for, in this case using FastMNN. We have now included a description of the process in the Extended methods section.

In the specific case mentioned by the reviewer, we sorted the ME-Transformed and MLL-ENL BM populations on the same day. We even went for a plate design, where each 96-well plate had cells of each type, to exclude plate effects. We are therefore confident that the separation mentioned by the reviewer is not a technical batch effect, but instead it is biological. In fact, these 2 cell types, although very similar, can be separated in representations that integrate multiple components when considering few neighbours (such as UMAP). This is not surprising since scRNA-Seq is a powerful technique capable to separate similar but not identical cells. Our rationale for using the PCA visualisation in this figure is to emphasise that: i) Parental and ME-Parental cells are extremely similar (cannot be separated by UMAP, see Figure S2B); ii) ME-Transformed and MLL-ENL BM are similar to each other, and when compared individually with the ME-Parental, they share the vast majority of differentially expressed genes.

Finally, as requested by the reviewer, we also included the information corresponding to the amount of variance associated with each PC in the plots before the FastMNN correction (see Figure S2A).

5. Figure 2C demonstrates projection of leukemic samples onto normal hematopoietic development. Additional details on how this projection was generated would be very helpful. Identifying all the colored populations explicitly would be very helpful (i.e., what is the pale

lavender population? What is the central red population?). Furthermore, the displays showing the projections of the parental and ME-parental are strikingly similar. It would seem that the probability that both populations would have many cells that project into basically the identical locations on this space is very low. Can the author comment on this significant concern, i.e. what is the probability that both populations what have scattered cells that project to almost identical positions in this space? Can the authors rule out some sort of bug or sample swap? If not, what is the biology underlying this observation?

As this point contains 3 subsections, we have labelled them a-c.

a) We have expanded the methods section to explain in more detail how the projections were performed (see Extended methods section). Projections of single cell transcriptomes into a large single cell transcriptional landscape are in our opinion a very powerful way to assess not only similarities/differences between cells, but also provide a very powerful way to map cells to their closest relatives in terms of differentiation status in an unbiased way.

b) The second point to this question is in relation to the populations within the landscape. These have been defined in our Dahlin et al Blood paper (2018) by Louvain clustering, as this is the currently accepted way of defining cell clusters in an unbiased/unsupervised way within scRNA-Seq datasets. Importantly, these rationally defined clusters will not map back in a 1-to-1 fashion to the conventional populations as defined by FACS. As shown in 2015 by Ido Amit's group, the problem here is not with the scRNA-Seq, but the FACS, because FACS populations are heterogeneous. About the identity of the "intermediate" clusters (such as the pale lavender), we feel it will not be helpful to come up with specific names for them, such as bipotent X+Y progenitors since within the hematopoiesis field, naming a population requires purification of the cells. But, since our cells are defined by computational means, we cannot purify their exact counterpart by FACS.

c) We were not surprised to see that the ME-Parental and Parental single cells mapped to very similar locations in the scRNA-Seq landscape. Although we agree with the reviewer that the projections are strikingly similar they are not identical (as also pointed out by reviewer #2 point 3). Importantly, all our other analysis (PCA and UMAP of scRNA-Seq as well as ATAC-Seq) shows that these cells are very similar. Indeed, on PCA and UMAP analysis, they intermingle! The reviewer also queried how we can be sure that there is no sample swap. Apart from the fact that we have been very careful with all our analysis anyway, we can provide added reassurance here, because we generated a custom genome-build for this study, which contains the MLL-ENL-GFP fusion gene sequence. Importantly, only cells labelled as ME-Parental had mapped reads against MLL-ENL, with none from cells labelled as Parental.

6. It is unclear why the authors decided to perform bulk ATAC-seq on just the parental and ME-parental cell lines. What was the experimental rational for thinking that chromatin accessibility would be different between the two populations? Is there evidence in the literature to suggest this? If the authors postulated that chromatin accessibility may play a role in early leukaemagenesis, then why was ATAC-seq not performed on the ME-preleukaemia and BM populations? Finally, given that single-cell RNA-seq was performed on the model system, why

did the authors choose to perform bulk ATAC-seq instead of single-cell ATAC-seq on the populations of interest?

As there are again several sub-questions, we provide our responses listed a-c below:

a) Why look at chromatin: The rationale to look at chromatin comes from the fact that MLL-ENL is known to have potential effects on chromatin. We have now amended the text to clarify this point. As it happens, we observed that as long as the cells are maintained in self-renewal conditions, there are no major effects on global chromatin accessibility patterns. This is not necessarily what we expected, as we knew the cells had different potentials (only the ME-Parental can transform to AML!). Since RNA-Seq is mostly concerned with mapping the “current” molecular state of a cell, chromatin analysis can also provide insights into the potential of a cell, e.g. what it will do (or will not do) in the future (e.g. if a gene locus is already accessible, it can be readily induced in future). We had therefore thought that looking at open chromatin might reveal some differences that could explain the difference in potential between ME-Parental and Parental. In this revised version of the article, we also included an additional time point to study the possible effects of the expression of MLL-ENL over time.

b) Why not ATAC-Seq on the ME-Transformed cells: we performed the ATAC-Seq experiments in the context of investigating the differences between Parental and ME-Parental cells. The ME-Transformed cells have a very different transcriptome to the Parental and ME-Parental cells. It is self-evident therefore that the chromatin patterns will also be different and we feel that additional mapping of open chromatin would not add majorly to our ability to achieve our goal. It may nevertheless be an interesting thing to do in future, just not in the context of this paper, where, as stated above, it would not add to the storyline.

c) Why not single cell ATAC-Seq: Firstly, single cell ATAC-Seq was very recent when this work was planned and it wasn't as accessible as currently. Secondly and more important, single cell ATAC-Seq is very sparse at the level of the individual cell. Therefore, most of the analysis ends up pooling cells into different groups, and then generating artificial “bulk” chromatin maps for these cell pools. We know from our scRNA-Seq that the Parental and ME-Parental cells do not show substantial heterogeneity. Doing the analysis at single cell level therefore would most likely end up in pooling everything back together afterwards anyway.

7. The ATAC-seq analysis is rather superficial, and thus it would be helpful to see a more detailed investigation/explanation into the ATAC-seq data (Figure 2Ei and 2Eii, Figure S2). The authors demonstrate approximately 10% unique peaks between the two populations – what are enriched in the two distinct peak sets? Are there any transcription factor motifs that are enriched in each population? It would likely be more helpful to see a MA plot rather than the plot shown in figure Eii.

We thank the reviewer for bringing up this point, because on reflection, we agree that it would have been good to include some more analysis on the ATAC-Seq data. For the revised manuscript, we have approached the analysis in a slightly different way. Instead of calling peaks and comparing called peaks between samples, we compared for each time point the coverage of all regions that had been called as a peak in any of the samples corresponding to

that time point. We feel that this analysis is less biased since it is independent of the threshold used to define peaks which usually varies between samples and therefore it is more in agreement with the current trend of analysis of these sort of samples. As mentioned before, we have also included an additional time point and, as the reviewer suggested, we have created new figures that contain MA plots. These results show that there are no statistically significant differences in accessibility between Parental and ME-Parental cells, at neither day 6 nor day 9 of culture in the presence of Flt3L and β -estradiol, with the exception of the DNA sequences corresponding to the MLL-ENL sequence introduced into the cells by retroviral transduction. Recovery of DNA corresponding to transduced viruses in chromatin analysis is a commonly seen artefact, presumably reflecting the increased copy number of those sequences in the transduced cells. We have also included visualisation tracks of the regions corresponding to MLL and ENL (see Figure S2).

Together, these additional results make the ATAC-Seq data a much more rounded part of the paper.

8. Can the authors comment on why the ME-preleukaemia population characterized in Figure 2 does not display the phenotypic heterogeneity observed after only 7 days exposure to IL-3 in figure 4.

We apologize, because we should have explained this issue in the text. The ME-preleukaemia cells have gone through a selection process of several weeks, consisting of serial replating assays and subsequent liquid culture, initially in the presence of SCF, IL-6 and IL-3, followed by gradual removal of SCF and IL-6. This selection results in a homogeneous population. By contrast, the Parental and ME-Parental cells in figure 4 have only been exposed to IL-3 for 7 days, and the heterogeneity here represents the way how individual cells respond to the initial change of conditions. We have amended the text to clarify these aspects.

Finally, the PCAs in figures 2 and 4 were computed independently, using different highly variable genes. Therefore, the genes that contribute to each component and the contribution of the common genes are not the same in the two figures, which account for the differences in heterogeneity of ME-Transformed cells between the 2 figures.

9. In Figure 4E, the authors show that MLL-ENL2 clusters with the parental, while MLL-ENL1 clusters with the ME-preleukaemia. How robust is this phenotypic divergence in the MLL-ENL populations? If this experiment was repeated, would you expect to see it again? When do this distinct populations emerge (3 days? 7 days?)? Do they persist? What is driving the divergence of the populations? Are they clonally related?

We are happy to say that this result is very robust, as we reproduced it. We have included single cell RNA-Seq of a biological replicate experiment as a new supplementary figure (Figure S4C). In this new replicate, as before, we can identify 2 populations of ME-Parental cells following 7 days of exposure to IL-3 that can be distinguished by the expression of *Elane* and *Cd74*. To answer the question about timing, we also now carried out scRNA-Seq analysis on separate days, which shows that the phenotypic divergence begins very early on, at around day 4,

however at this time point it is difficult to differentiate these cells from more undifferentiated Parental cells (also included in new supplementary Figure S4C). This timecourse analysis also allowed us to show that the heterogeneity does persist, however prolonged exposure selects out a more uniform population from day 11 (also in same supplementary figure S4C). We believe that the selection (and thus divergence between Parental and transduced) is driven by the fact that only the population that acquires granulocyte features is able to proliferate since in MLL-ENL driven AML there is a block in the maturation in the myeloid compartments and cells acquire a granulocyte phenotype.

And in terms of the final and interesting sub-question about clonal relationship, we understand that the referee's question is in relation to the clonality of the subpopulation MLL-ENL1. As already stated in the response to the editor's comments, the Parental cells are clonal. We believe that the most likely explanation for the heterogeneity in the response is that even within a clonal population, individual cells will exist in distinct molecular states, where processes such as cell cycle can have a major influence on how cells will be able to respond to external stimuli.

10. It is not entirely surprising that a drop-out screen in a rapidly proliferating cell line would result in hits to DNA-repair and cell cycle genes. Were there any hits that were surprising (i.e., not involved in DNA repair, cell cycle, glycolytic metabolism)? These might prove to be a more attractive target for potential therapeutics.

We are grateful that the reviewer is bringing up this point, because on reflection, we should have elaborated on this. It is important to note that the Parental cells grow faster than the ME-Transformed cells, yet the DNA repair and cell cycle genes identified in our paper were specific dropouts in the ME-Transformed cells, and not in the Parental. They were also upregulated in the expression analysis. We interpret this as indicating that the ME-Transformed cells have adopted a cellular state where cell division (proliferation) is actually under some strain, counterbalanced by pushes to differentiate and/or die. The upregulation of genes such as ATM is required for the cells to "get by" under these stressful conditions. This model is consistent with our observation that even though the ME-Transformed cells express higher levels of these genes, they are nevertheless more sensitive to the inhibitors. The idea then is that the pathways are of more critical relevance, and already operating near their limit in the ME-Transformed cells. We have amended the text to make this important point (see page 21).

Specific dropout genes for ME-Transformed cells included many putative interesting targets, including a category of genes involved in "chromatin organisation" (p-adjusted value 2.59E-17). Therefore, by no means all of the candidate targets identified in our study are thought to be involved in cell cycle, DNA repair or glycolysis. We have now included this category in the main text to enhance this important point.

11. A figure demonstrating the overall model would be helpful to reference in the discussion.

Panel A) in Figure 1 contained an outline of our experimental model. We have amended the figure legend, so that it is now suitable to be referred to in the discussion section when we talk about the aspects that set our model apart from previous approaches.

Reviewer#2: (Expertise: Crispr screen, Remarks to the Author)

We were pleased to read that this reviewer agrees that our “*experimental system using Hoxb8-FL allows for more accurate characterization of molecular events during the early phase of leukemogenesis, which have been difficult to analyze because of sample availability and cell heterogeneity*”. The reviewer also made a number of specific comments which he/she felt needed to be addressed to make our paper suitable for publication. We have addressed all these points as outlined below:

1. The authors supplied a list of CRISPR screen hits (dropout genes) but no other essential datasets (raw gRNA counts for each cell line, each replicate and each time point, and statistical outputs). Based on the number of dropout genes, the screens seemed to be conducted successfully; however, due to the lack of the dataset, I cannot assess these screens further. All datasets are valuable resource for the community as well. The authors must make the full set of the datasets available.

We agree entirely with the sentiment expressed by the reviewer. We have now made available the raw sequencing results and have included new supplementary material that contains the statistical results per gene, datapoint and sample (Table S3).

2. As validation of the screen hits, the authors performed pharmacological inhibition of 3 selected genes. These must be first performed by a genetic means (ie. by gRNA). CDC7i and LDHAi showed very high IC50 and very small difference on sensitivity between preleukemic and parental cells; therefore, it does not support the claim that these genes are specifically required in pre-leukemic cells.

We have now performed the validation experiments requested by the reviewer, which are included as Figure S5. These experiments show that the CDC7 and LDHA are indeed valid therapeutic targets, and that investment into the development of more potent inhibitors may well be a worthwhile avenue for future investigation.

3. Fig. 2C. The plots for Parental and ME-Parental are extremely similar (but does not look completely identical). Are these shown with appropriate datasets? Based on the 'Parental' plot, it seems to me that this population is quite diverged and shows considerable heterogeneity. It would be useful to analyze the scRNA-seq data from the Parental cells more deeply to understand the heterogeneity that exists in Hoxb8 cells and discuss any detrimental effects/consequences for the system developed here.

As already alluded to in our response to reviewer #1, point 5, the Parental and ME-Parental samples are very similar at the molecular level when investigated either by PCA, UMAP, projection or ATAC-Seq, yet we can be sure that there is no sample swap, because we have used a custom genome build to map the sequencing reads, and therefore can confirm that reads mapping to the MLL-ENL fusion gene are only seen in the ME-Parental sample. As to the question about the source of heterogeneity on the scRNA-Seq of the Parental cells, this is a very interesting point, which we believe is most likely related to the fact that they are an LMPP-

like population, and their transcriptomes therefore show similarity with progenitors both in the lymphoid and myeloid territory of our single cell transcriptome landscape. To investigate this issue more formally, we have now obtained UMAPs (Figure S2B) where we cannot detect separation or subgroups within these populations. We have also amended the text of our revised manuscript to clarify this issue (see page 9).

Minor comment:

Cell labeling is not accurate enough to specify which samples were used. For instance, "Parental" is used a control, but is this always MSCVGFP transduced? How long was ME-Parental cell line cultured before each analysis? Is this cell line stable and does not show any spontaneous transformation?

We apologize for any confusion that may have arisen from imprecise wording. Throughout the paper, whenever we use the term "Parental", this refers to cells transduced with the negative control MSCV-GFP vector (we have gone through the text carefully to clarify this throughout). For all experiments described in the manuscript, ME-Parental cells were cultured for a total of 9 days in the presence of Flt3L and β -estradiol following MLL-ENL transduction. A fresh batch of cells was used for each experiment, thawed approximately 10 days before the start of the experiment, then transduced with the vector containing the MLL-ENL translocation, GFP-sorted at day 6 and then expanded for 3 more days (we have amended the methods to clarify this point). As to the final query, the ME-Parental cells are stable in culture. We have now performed growth curve analysis, demonstrating that the cells can grow for over a month without changing their growth rate, and also maintaining stable expression of cKit (analysed by flow cytometry). Of note, we have also performed scRNA-Seq following 12, 16 and 20 days of culture in the presence of Flt3L and β -estradiol and we do not find differences at the transcriptomic level but we do not include these data in our article due to the low number of cells analysed. Additionally, we have been able to obtain ME-Transformed cells from ME-Parental cells cultured for 6, 9, 12, 16 and 20 days in the presence of Flt3L and β -estradiol, which are phenotypically like the ones described in the main text. We never observed spontaneous transformation.

Reviewer #3 (Expertise: Hematopoiesis and AML models, Remarks to the Author):

We were delighted to read that this reviewer identified strong points in our paper such as "*the development of a cell line system that could in theory be used to study clonal succession in AML driven by a variety of mutations or gene rearrangements, and that is amenable to high-throughput screening technologies*" and the "*identification of some novel potential therapeutic targets that could be further tested in MLL-rearranged AML*".

The reviewer also identified a number of points which he/she felt needed to be addressed, which we have done as outlined below:

Major:

1. Unlike some AML mutations and chromosomal rearrangements, MLL-ENL AML has not been demonstrated to have a pre-leukemic phase in patients. Therefore, it is unclear why the authors are calling these "pre-leukemic" cells, and why they chose MLL-ENL to model the pre-leukemic state. Therefore, the clinical significance of this model system is unclear.

Modelling of human AML in mice has certainly limitations. The modelling of leukaemias involving MLL rearrangements has been under investigation for a long time and the nature of the leukaemia initiating cell is still a point of debate. Thus, models described up to now are suffering from either limited numbers of starting cells or heterogeneous mixtures of starting cells. The comparison and analysis of the early events of transformation are difficult to study using these models. Our model circumvents these limitations since the starting population of cells is identified, there is no limit in starting numbers and the process can be deployed at will. Our model therefore provides a great opportunity to study early changes in the “transformation” process towards a leukaemic cell.

We do agree with the referee that MLL-ENL has not been demonstrated to have a pre-leukaemic phase in patients. However, *in vitro* and *in vivo* studies have shown the requirement of additional factors for transformation of cells carrying MLL rearrangements. In our case, the “transformed” cells are able to generate leukaemia *in vivo* with great penetrance. Because of this potential, we called them “preleukaemic” but we agree with the reviewer that the terminology may induce confusion and therefore we suggest the term “ME-Transformed” for these cells, since they have been through a “transformation” process to acquire leukaemic potential. We have amended the entire manuscript in light of this revised nomenclature.

2. The parental Hoxb8-FL cell line that is used as the negative control cannot be compared to healthy hematopoietic cells because it is immortalized with Hoxb8. Due to this baseline immortalization, even the parental cells transduced with empty vector could be thought of as "pre-leukemic", and are in no way normal.

As already stated in our response to the editor’s points, Hoxb8-FL cells are not an immortalized cell line. The cells have a conditional differentiation block, mediated by inducible Hoxb8 expression and requiring supplementation with the cytokine Flt3 ligand. This allows the maintenance of the cells as a cytokine-dependent and self-renewing LMPP pool. Importantly, when Hoxb8 induction is removed, the cells have multilineage differentiation capacity. When transplanted *in vivo*, they behave just like LMPPs, which is that they provide short-term contribution to myeloid and lymphoid lineages. Critically, they are not leukaemogenic. The original paper showed the short-term *in vivo* contribution. We have repeated these experiments, and now include data for this in the revised paper (Figure S1) together with a figure for the reviewers (see Reviewers’ Figure R1). Neither the original paper nor our own repeat of these *in vivo* transplant experiments saw any evidence for leukaemia development.

The reviewer is correct in querying how “normal” the Hoxb8-FL cells are. When cultured *in vitro*, we show that their transcriptomes are very similar to cells occupying the “LMPP-

territory” within a single cell transcriptional landscape derived from primary bone marrow cells. Moreover, when differentiated *in vitro* or transplanted *in vivo*, they are very similar to primary cells in terms of their functional behaviour. We would argue therefore that the cells provide a very attractive model for primary multipotent progenitors, while it is clear of course that the “trick” used to let them self-renew *in vitro* (conditional supply of Hoxb8 + Flt3 ligand) does introduce a potential deviation from the molecular and cellular state of normal primary cells. We have amended the manuscript to clarify this issue.

At the end of the day, the Hoxb8-FL system is a model system, and we believe it is a very powerful model system. Modern biomedical research constantly requires new model systems to drive advances. We show here that the Hoxb8-FL system provides for the first time a clear-cut “before/after” scenario for studying AML development, because (i) we know the exact nature of the cells that are transduced with the MLL fusion transgene, and (ii) we can propagate these cells *in vitro*. Previously, researchers used heterogeneous populations such as lineage-negative bone marrow cells. The heterogeneous nature of these cells obscures a direct knowledge of which exact cell was actually transformed. Moreover, lineage-negative cells cannot be propagated *in vitro*, thus making it impossible to perform dropout screens. This is indeed one of the key advances of our approach, because for the first time, we have the opportunity to compare molecularly (gene expression) and functionally (dropout screen) the target cell of transformation with the end result of transformation (e.g. the true “before/after” comparison).

3. In Figure 1, the authors claim that the "ME-pre-leukemic" cells cause leukemia when transplanted into mice, but not the "parental" cells. However, only survival curves and organ weights are shown. To make the diagnosis of AML in these mice, it would be necessary to include histology images of blood, or of bone marrow, spleen and/or liver sections. In addition, flow cytometry analysis of the leukemic cells from the mice should be shown, because these cells could change their cell surface markers in vivo.

This same information was requested by reviewer #1, point 1. We agree that these are all very good points in demonstrating the validity of a leukaemia mouse model. We have therefore now performed histological analysis of organs from leukaemic mice for ME-Transformed transplant experiments (see Figure S1C). Additionally, we have also included the characterisation of leukaemic cells from 3 different leukaemic mice by flow cytometry (see Figure S1D).

4. In Figure 1, transplantation of the "ME-parental" cells is not demonstrated, and it is not clear whether these cells are also capable of causing leukemia. This is critical to the interpretation of the sc RNAseq studies presented later, in which the ME-parental cells are compared with "ME-preleukemic" cells.

Our concerted analysis of the ME-Parental cells shows that they can be coaxed into a leukaemic state, but this requires that they are relieved from the Hoxb8/Flt3 ligand self-renewal program. While maintained in self-renewal conditions, the ME-Parental cells maintain their molecular state, and also do not change their growth conditions or surface marker profile (see new data

generated in response to reviewer #2 – minor comment). Given that the ME-Parental cells are virtually identical to the Parental condition, we believe that a comparison between ME-Parental and ME-Transformed is valid.

In terms of leukaemogenic potential, it is well known that ckit-positive cells can produce leukaemia *in vivo* when transduced with MLL-r oncogenes, without undergoing the transformation process *in vitro* (Basheer et al. 2019. J Exp Med. Apr 1;216(4):966-981. doi: 10.1084/jem.20181276). ME-Parental cells may therefore have leukaemogenic potential when transplanted directly into mice. In this setting, the recipient would provide the adequate niche and signals to produce the transformation *in vivo*. Importantly, the resulting cells would not be ME-Parental anymore and the monitoring of the process would not be accessible.

Minor:

1. The generation of "ME-parental" cells needs to be better defined. It is not clear from the results section how these cells were generated, and how they are different from "ME-preleukemic" cells.

We have expanded both the results and methods sections and the relevant figure legends to clarify this point.

2. In Figure 3, only representative flow plots are shown, but there is no statistical analysis included, and N=2. This experiment should be repeated at least 3 times, and statistics need to be shown for all of the populations in question.

Figure 3A now includes the mean for 3 independent experiments and we have included a supplementary figure (Figure S3A) to show variability and statistics.

3. In the table in Figure 3D, no confidence intervals or p values are shown.

Again, we have increased the number of replicates for this figure and we have redesign the figure to include statistical analysis.

4. In the sc RNAseq analysis shown in Figure 4, it is interesting that the MLL-ENL cells were separated into two populations, MLL-ENL1 and MLL-ENL2. However, the biological significance of these two populations is unclear. It would be informative to sort and transplant each of these populations into mice, and determine which one is more efficient at initiating leukemia in vivo.

Reviewer #1 (point 9) also asked about the MLL-ENL1 and MLL-ENL2 populations. As stated in the response to reviewer #1, this observation was highly reproducible. We believe that MLL-ENL2 is associated with differentiation, and therefore highly unlikely to be the source of serial replating activity while MLL-ENL1 is similar to transformed cells. Sorting of populations defined from scRNA-Seq is not trivial, because we would first need to identify and validate a

surface marker combination that can cleanly separate the two populations. Moreover, our experimental approach did not entail direct transplantation of cells at this stage, but only following the serial replating *in vitro*. We would therefore argue that the transplantation experiments referred to by the reviewer would be more appropriate for a follow-up study, also considering that this point was under the “Minor” heading.

5. While it is encouraging that some of the candidates from the CRISPR dropout screen could be pharmacologically validated, only the ATM inhibitor appears to be active in these cells at doses that may be achievable in patients. If this drug could be validated in a mouse model of MLL-rearranged AML, this would increase the clinical relevance of the study.

We had clearly cited in our paper that the ATM inhibitor used by us has already been validated in a mouse model *in vivo* (Morgada-Palacin et al, Science Signalling 2016). It therefore served as a positive control, as stated in our original submission. Importantly, we have now experimentally validated all three hits with gene-specific guide RNAs, thus providing genetic evidence supporting the inhibitor studies.

Figure R1.-

Lethally irradiated mice were transplanted with either Parental (n=5) or ME-Transformed cells (n=5), together with total bone marrow cells as helpers. Peripheral blood samples were taken after transplantation and assessed for chimerism by flow cytometry. Plots show the expression of GFP (day 7) and the distribution of Cd45.1 and Cd45.2 (days 14, 21, 31 and 68) for all mice in the experiment at all assessed time points. Recipient animals and helper cells were Cd45.2-positive and GFP-negative while Parental and ME-Transformed cells were Cd45.1-positive and GFP-positive.

REVIEWERS' COMMENTS:

Reviewer#1: (Remarks to the Author)

Basilico et al. have addressed many of our initial concerns in their revised manuscript, including providing clarification on experimental rationale and data as well as providing some additional analyses of the ATAC-seq and scRNA-seq data as requested. While we have some remaining minor concerns (outlined below), we think that overall, the manuscript is suitable for publication in Nature Communications.

1. The authors suggest that the two distinct MLL-ENL populations (MLL-ENL1 and MLL-ENL2) that form after IL-3 differentiation for 7 days (Figure 4B) most likely reflect heterogeneity in the response of individual cells to differentiation conditions. However, it is worth highlighting in the text that the scRNA-seq data shown in this panel for GFP and MLL-ENL shows a clear trend towards higher expression of the transgene in the MLL-ENL1 population, despite not reaching statistical significance (perhaps due to insufficient number of cells assayed). This would be consistent with the greater molecular changes in this population relative to the MLL-ENL2 population.

2. In certain experiments, it is unclear whether differences in the ME-transformed vs parental lines are due to MLL-ENL transformation or differences in culture conditions (IL-3 for ME-transformed vs FLT3L/b-estradiol for parental lines). This is especially of concern for interpreting the results of the CRISPR screen for genetic vulnerabilities unique to the ME-transformed line. In fact, the authors point out that FLT3 and IL3RA were unique hits in the parental and transformed lines, respectively. To help separate these effects, the authors could remove hits that overlap with transcriptional changes associated with transiently culturing the parental cells in IL-3 conditions.

Reviewer#2: (Remarks to the Author)

The authors have sufficiently addressed my initial comments. I am happy to support the publication of the manuscript.

Reviewer#3: (Remarks to the Author)

The authors have been very responsive to all of the reviewers' comments, and the manuscript is substantially improved.

REVIEWERS' COMMENTS:

Reviewer#1: (Remarks to the Author)

Basilico et al. have addressed many of our initial concerns in their revised manuscript, including providing clarification on experimental rationale and data as well as providing some additional analyses of the ATAC-seq and scRNA-seq data as requested. While we have some remaining minor concerns (outlined below), we think that overall, the manuscript is suitable for publication in Nature Communications.

1. The authors suggest that the two distinct MLL-ENL populations (MLL-ENL1 and MLL-ENL2) that form after IL-3 differentiation for 7 days (Figure 4B) most likely reflect heterogeneity in the response of individual cells to differentiation conditions. However, it is worth highlighting in the text that the scRNA-seq data shown in this panel for GFP and MLL-ENL shows a clear trend towards higher expression of the transgene in the MLL-ENL1 population, despite not reaching statistical significance (perhaps due to insufficient number of cells assayed). This would be consistent with the greater molecular changes in this population relative to the MLL-ENL2 population.

We have included a sentence in the results section which covers the point referred to here by the reviewer (page 11, line 247).

2. In certain experiments, it is unclear whether differences in the ME-transformed vs parental lines are due to MLL-ENL transformation or differences in culture conditions (IL-3 for ME-transformed vs FLT3L/ β -estradiol for parental lines). This is especially of concern for interpreting the results of the CRISPR screen for genetic vulnerabilities unique to the ME-transformed line. In fact, the authors point out that FLT3 and IL3RA were unique hits in the parental and transformed lines, respectively. To help separate these effects, the authors could remove hits that overlap with transcriptional changes associated with transiently culturing the parental cells in IL-3 conditions.

To answer the reviewer's comment, we obtained the differentially upregulated genes in Parental cells cultured in the presence of IL-3 for 7 days compared to the Parental cells cultured in the presence of Hoxb8 and β -estradiol. We then compared this list with the list of 127 dropout genes that are unique to the ME-Transformed line and are also upregulated specifically in MLL-ENL1 cells. There was no overlap between the 2 lists.

We have included a sentence in the main manuscript to state this relevant point (page 15, line 342).

Reviewer#2: (Remarks to the Author)

The authors have sufficiently addressed my initial comments. I am happy to support the publication of the manuscript.

Reviewer#3: (Remarks to the Author)

The authors have been very responsive to all of the reviewers' comments, and the manuscript is substantially improved.